



# ReOBS: a new approach to synthetize long-term multi-variable dataset and application to the SIRTA supersite

Marjolaine Chiriaco[1], Jean-Charles Dupont[2], Sophie Bastin[1], Jordi Badosa [4], Julio Lopez[2], Martial Haeffelin[2], Helene Chepfer [3], Rodrigo Guzman[3]

(1) LATMOS/IPSL, UVSQ Université Paris-Saclay, UPMC Univ. Paris 06, CNRS, Guyancourt, France
(2) Institute Pierre Simon Laplace, Ecole Polytechnique, Université Paris-Saclay, Palaiseau, France.
(3) Laboratoire de Météorologie Dynamique, Univ. Pierre and Marie Curie, Paris, France.
(4) Laboratoire de Météorologie Dynamique, Ecole Polytechnique, Palaiseau, France

*DOI:* http://dx.doi.org/10.14768/4F63BAD4-E6AF-4101-AD5A-61D4A34620DE

Or 10.14768/4F63BAD4-E6AF-4101-AD5A-61D4A34620DE at dx.doi.org

*Dataset available at:* *http://sirta.ipsl.fr/reobs.html* (tab download, no password required)





**Abstract**
A scientific approach is presented to aggregate and harmonize a set of sixty geophysical
variables at hourly scale over a decade, and to allow multiannual and multi-variables
studies combining atmospheric dynamics and thermodynamics, radiation, clouds and
aerosols, from ground-based observations. Many datasets from ground-based
observations are currently in use worldwide. They are very valuable because they
contain complete and precise information due to their spatio-temporal co-localization
over more than a decade. These dataset, in particular the synergy between different type
ob observations, are under-used because of their complexity and diversity due to
calibration, quality control, treatment, format, temporal averaging, metadata, etc. Two
main results are presented in this article: (1) a set of methods available for the
community to robustly and reliably process ground-based data at a hourly time scale
over a decade is described, and (2) a single netCDF file is provided based on the SIRTA
supersite observations. This file contains approximately sixty geophysical variables
(atmospheric and in-ground) hourly averaged over a decade for the longest variables.
The netCDF file is available and easy to use for the community. In this article,
observations are "re-analyzed". The prefix "re" refers to six main steps: calibration,
quality control, treatment, hourly averaging, homogenization of the formats and
associated metadata, and expertise on more than ten years of observations. In contrast,
previous studies (i) took only some of these six steps into account for each variable, (ii)
did not aggregate all variables together in a single file, and (iii) did not offer an hourly
resolution for about sixty variables over a decade (for the longest variables). The
approach described in this article can be applied to different supersites and to additional
variables. The main implication of this work is that complex atmospheric observations



are made readily available for scientists that are non-experts in measurements. Dataset
from SIRTA observations can be downloaded on http://sirta.ipsl.fr/reobs.html (tab
download, no password required) under DOI http://dx.doi.org/10.14768/4F63BAD4-
E6AF-4101-AD5A-61D4A34620DE.



## 1. Introduction

The Intergovernmental Panel on Climate Change (IPCC) simulations show a large spread between models when predicting future climate at global scale, but also when representing the observed current climate. These model uncertainties are larger at the regional scale and at short time scale (e.g. seasonal scale). These scales are however key for the impacts assessment. For example models do not reproduce observed magnitudes of interannual and seasonal variability and extremes in temperature and precipitation (Terray and Boé, 2013). Hawkins and Sutton (2009) also show that climate natural variability is the main source of uncertainty to predict regional climate evolution at the scale of 10-20 years (compared to the selected scenario or model). Observations of the atmosphere must be considered in order to improve both our knowledge of the processes that create this temporal variability and the simulations uncertainties. These observations must describe atmospheric processes that involve a large number of variables in the atmospheric columns and in the ground, and at various spatial and temporal scales.

Multiannual and multi-variables datasets are therefore necessary. Many of these datasets from ground-based observations have a significant scientific value because they contain complete and precise information on one or several decades, due to their spatio-temporal co-localization. Supersite observatories such as the Site Instrumental de Recherche par Télédétection Atmosphérqiue (SIRTA, Haeffelin et al. 2005) or the different Atmospheric Radiation Measurements (ARM, Ackerman et al. 2003) are among these sets of observations. But they are under-used, in particular the observation synergy aspects, because of their complexity and diversity in terms of calibration procedures, quality control, data treatment, file format, temporal representativeness,



metadata etc, because of the weak magnitude of the signals to be highlighted (e.g. trend
versus natural variability), and because of the complex connections between local-scale
processes and climatic-scale anomalies (e.g. links between ground/boundary
layer/atmosphere processes and heatwaves, as in Chiriaco et al. 2014).
An important homogenization work was needed on these observations. Homogenization
has been performed for ARM observatories leading to the ARMBE (ARM Best Estimate)
data product (Xie et al. 2010), which is the "ARM datastreams specifically tailored to
climate modelers for use in the evaluation of global climate models. They contain a best
estimate of several cloud, radiation, and atmospheric quantities. The ARMBE dataset
was    created    to    showcase    all    the    flagship    products    of    ARM"    (from
https://www.arm.gov/capabilities/vaps/armbe). A specificity of ARMBE products is
that all variables are gathered in only two files: ARMBEATM (ATM for atmosphere) for
many atmospheric state profiles and surface quantities, and ARMBECLDRAD (CLDRAD
for cloud and radiation) that contains a best estimate of several selected ARM and
satellite-measured cloud and radiation relevant quantities.
In this article, additional steps are applied to the observations and precisely described in
order to understand how the observations are "re-analyzed". This method is called
ReOBS. The prefix "re" refers to six main steps on more than ten years of observations:
calibration, quality control, algorithmic treatment, hourly averaging, homogenization of
the data formats and associated metadata, and scientist expertise. In contrast, previous
studies (i) only take into account some of these six steps for each variable, (ii) do not
aggregate together all variables in a single file, (iii) do not offer an hourly resolution for
about 60 variables on a decade (for the oldest variables).
The ReOBS method was initially inspired by the ARMBE project and has been developed
at SIRTA (located 20 km southwest of Paris, France). The SIRTA observatory has been



collecting data for fifteen years from active and passive remote-sensing, in situ
measurements at the surface, in the ground, and in the planetary boundary layer. Early
versions of SIRTA ReOBS dataset ("Re" stands for different steps of re-processing, see
next sections, and "OBS" stands for observations) have already been used in scientific
studies that required the multi-variables and multi-temporal scales available in the
SIRTA-ReOBS dataset (Cheruy et al. 2012, Chiriaco et al. 2014, Pal and Haeffelin 2015,
Bastin et al. 2016, Dione et al. 2016). The ReOBS method has also been tested for other
supersites for some variables (classical meteorology, radiative fluxes, heat fluxes):
Cabauw (in Netherlands) and Chilbolton (in England) supersites in the framework of
EUCLIPSE European project (European Union Cloud Intercomparison, Process Study
and Evaluation project), and CO-PDD (Cézeaux – Opme – Puy De Dôme at Clermont
Ferrand in France) and P2OA (Plateforme Pyrénéenne d'Observations Atmosphériques
at Lannemezan in France) in Dione et al. (2016).
The objective of the current paper is to present a scientific approach (ReOBS) to
aggregate and harmonize about fifty geophysical variables at hourly scale on a decade,
to study atmospheric dynamics and thermodynamics, radiation, clouds and aerosols,
from ground-based observations. This paper presents two main results: (1) a set of
methods available for the community to process ground-based data robustly and
reliably at an hourly time scale over a decade; (2) provision of a single netCDF file
containing about fifty substantial geophysical variables hourly averaged over a decade
for the oldest ones, easily usable for the community.
The SIRTA observations used for applying the ReOBS method are described in Section 2.
The method used for ReOBS is then detailed in Section 3. Section 4 presents the contents
of SIRTA-ReOBS file and its major strengths: the vertical profiles, the multi-temporal



scales, and the multi-parameter specificity. Discussion and conclusions are drawn in
Section 5.

2. Observations

2.1. SIRTA observatory

SIRTA is a French national observatory dedicated to the monitoring of tropospheric
clouds and aerosols, the dynamics and thermodynamics of the boundary layer, and the
turbulent and organized transport of water and energy near the surface. The SIRTA
observatory is a mid-latitude site (48.71°N, 2.2°E) located in a semi-urban area, on the
Saclay plateau 20 km southwest of Paris, and hosts active and passive remote sensing
instruments since 2002 (Haeffelin et al. 2005). The SIRTA missions are 1) to monitor
continuously and on the long-term the atmospheric column using a core ensemble of
instruments, 2) to coordinate field campaigns in order to address specific scientific
questions, such as processes related to water vapor and clouds, the ultraviolet radiation
or the aerosol physics and chemistry, and 3) to provide teaching resources and to host
experimental training activities.
Figure 1 shows a selection of SIRTA routine measurements from the different on-site
locations (e.g. roof, mast, plain). The measurements used in the current study are listed
in Table 1. Lidars play a special role in the SIRTA instrumental park because several
lidars have been deployed at the SIRTA Observatory over the past 15 years, providing a
unique 3D-database: (1) a dual-wavelength (532 and 1064 nm) depolarization lidar
(called LNA for "Cloud and Aerosol Lidar", used in the current study) from 2002 until
2015 [Haeffelin et al., 2005], (2) a multi-wavelength elastic (355, 532, 1064 nm) and
Raman (387, 408, 607 nm) depolarization lidar (called IPRAL for "IPSL Hi-Performance
multi-wavelength Raman Lidar for Cloud Aerosol Water Vapor Research") since mid-





2015, (3) an automatic 355 nm backscatter and depolarization lidar (Leophere ALS450,
used in this study) from 2008 until 2014, and (4) an automatic 1064 nm Lidar
ceilometer (Lufft CHM15k) since mid-2015. The different lidars differ significantly in
complexity, emitted power, detection channels, signal-to-noise ratio, and capacity to
operate autonomously. For instance the LNA backscattered signal provides information
on the presence of clouds and aerosols in the vertical column between 0.5 and 15 km
altitude whereas the ALS450 backscatter lidar signal is exploited between 0.2 and 10
km.

2.2. SIRTA measurements used as inputs for ReOBS

Table 1 shows the measurements used as inputs to create the SIRTA-ReOBS file. The
table contains the instruments name, the physical bounds of the measurements, the
native resolution of the measurements, and the available period of observation. This set
of variables includes *in situ* measurements ((1- 6) and (11-14) in Table 1), passive
remote-sensing measurements ((7-10) and (17-20)), and active remote-sensing
measurements ((15-16) in Table 1).
These different measurements are used to create the geophysical variables listed in
Table 2. Some of the geophysical variables are directly measured, and some others
require advanced data processing, such as substantial quality control or algorithm
application. Data processing performed independently of the ReOBS processing chain
and already published is described and referenced in Table 2. The data processing
developed in the framework of the ReOBS project is described in Section 3.
In the rest of the article, the geophysical variables are split into four groups. Group A
contains the *standard meteorology variables* (first block in Table 2) such as 2-m
temperature, pressure, wind speed and direction, relative humidity, etc. Group B



contains the *advanced non-standard meteorology variables* (second block in Table 2)
such as radiative fluxes, heat fluxes, in-ground temperature and moisture, etc. These
latter variables are directly measured but are usually not available from typical weather
stations because they require advanced technologies, for instance based on remote-
sensing. Group C contains *variables retrieved from measurements* using algorithms
applied to remote-sensing measurements (third block in Table 2) such as cloud fraction,
water vapor content, etc. Finally, group D contains atmospheric vertical profiles from
lidar (fourth block in Table 2).

3. The ReOBS method

3.1. ReOBS general processing chain

The 14 year-long SIRTA-ReOBS dataset is contained in a single *netCDF* file containing
hourly values of 63 physical variables listed in Table 2. The short and standard name
used for each variable in the ReOBS dataset follows the Coupled Model Intercomparison
Project (CMIP) and the Climate Forecast (CF) conventions, respectively, when available.
For variables not included in CMIP or CF conventions, classical names are used or new
ones are created.
The strength of the ReOBS dataset is that all variables are processed using the same
high-level processing chain, completed by some sub-processing computations specific to
each variable. Figure 2 shows the ReOBS processing chain (in blue on fig. 2), which
starts after the acquisition process (in orange on fig. 2). Steps outside of the ReOBS
processing chain are marked in green.
For each variable (lidar profiles excepted), the hourly mean values are calculated from
the native resolution data (5 s to 1 min) by averaging all the data available within +/- 30



min around the full hour in order to be consistent with outputs from Global Circulation
Models (GCM) and Regional Climate Models (RCM). Each hourly variable is completed by
its intra-hour standard deviation. The hourly standard deviation of each variable helps
in detecting non-physical spikes (i.e. successive increase and decrease) and dips (i.e.
successive decrease and increase in the signal). This temporal variability information is
also useful to document large changes in the atmospheric conditions such as a cold front
for air temperature, broken clouds for radiative fluxes, and summer storms for
precipitations or latent heat fluxes.
Variables entering the ReOBS dataset are quality-controlled at their native time
resolution. The quality control test consists in verifying that the variable lies within
physical bounds (Tab. 1). Calculations of hourly mean and standard deviation only use
native resolution data that have passed quality control. A simple informative quality flag
is associated to each hourly value of a variable:

216         - 0: quality control is OK

217         - 1: there are valid data but for less than 50% of the period (that is, for less

than 30 minutes)

- there is no flag 2 because it is used for internal control

220         - 3: data is unavailable for the entire hour (no measurements or less than

50% of the measurements in the hour passes the quality control). In this case, the
hourly value is set by convention to -999.96.

Beside the systematic quality tests described above, some additional complementary
quality controls have been applied to specific variables, as described in the following
subsections (Sect. 3.2 to 3.5).




### 3.2. Specific computations for standard meteorological variables

Classical meteorological variables collected at three different locations are included in
the ReOBS dataset: 1) the first group of variables is collected at the supersite SIRTA and
has the advantage of being representative of the very local meteorology since the
beginning of the supersite activities, 2) the second group of variables aims at
characterizing the surrounding meteorology around the SIRTA site, and 3) the third
group of variables is from the standardized Météo-France station, collected at Trappes,
15 km away from the SIRTA supersite. These three different datasets are identified with
the suffixes -SIR, -REG, -TRP respectively in the following.

(i) Description of surrounding meteorology around SIRTA site.
Figures 3b, 3c, and 3d illustrate the difference in the air temperature, wind speed, and
cumulated precipitations at three Météo-France stations within a 50x50km domain
around the SIRTA supersite: in Trappes (48.8°N, 2.0°W), in Paris-Montsouris (48.8°N,
2.3°W) and in Orly (48.7°N 2.4°W).
The Probability Density Function (PDF) of the 2-m air temperature (noted *tas* in SIRTA-
ReOBS) shows an offset of about 2°C for Paris-Montsouris site compared to the Orly,
Trappes and SIRTA sites, which is due to the urban heat. Maximum values for the mean
wind speed value (noted *sfcWind* in SIRTA-ReOBS) are measured at the Orly site and the
mean wind speed is around 3 m.s$^{-1}$ at SIRTA. Note that measurements at the SIRTA site
are performed over a roof: wind speed is thus measured at 10 m above the roof,
corresponding to 25 m above ground level, whereas it is measured at 10 m above
ground level for the other stations. Even if a ground level standard (Météo-France-like)





meteorological station is also present at SIRTA, the rooftop measurements were
preferred for the ReOBS file because they started earlier (in 2003) than the standard
meteorological station (in 2006). The four stations are characterized with a cumulated
annual precipitation ranging between 600 and 700 mm/year.
The data collected in the three stations around the SIRTA supersite are used to
characterize the surrounding 2-m meteorology. A weight is associated to each station
based on the following method: the 50 km x 50 km domain is divided by a factor 300 x
300, leading to $90.10^3$ grid boxes, and each site is given a weight inside the 300x300 box
region, which corresponds to its geometric representativeness. The weight of the
Trappes station is then 44.4%, the weight of the Orly station is 34.5% and the weight of
the Paris-Montsouris station is 21.1% (Fig. 3a). The regional scale meteorology variables
v (-REG) included in ReOBS are then obtained from:

(1)

$$\bar{v} = \sum_{i}^{n} x_i w_i$$

where x = {$x_1$, …, $x_4$} is the set of values taken by a variable v (2-m temperature,
humidity, wind) at each of the four stations, and w = {$w_1$, …, $w_4$} is the station weight.

(ii) Quality control of the standard meteorological variables
The quality control for meteorological variables listed in tab. 2 consists of two additional
tests compared to what was indicated in Sect. 3.1. The goal of the quality control is to
reject unphysical values and to reject values with unrealistic temporal variability
(Tables 1 and 3), e.g. non-physical jump in the data record, non-physical persistence in
time of the measured values.



Non-physical jumps in the data are detected at native high temporal resolution as the
correlation between the adjacent samples increases with the sampling rate. If the
difference between two successive measurements is more than a specified limit given in
Tab. 3 (these tests are about to be refined in a new study that will give a new version of
the SIRTA-ReOBS file) the current measurement is rejected but it is used for checking
the temporal consistency with the next measurement. Two examples of measurements
that did not pass the quality control tests are  shown in Fig. 4a and 4b for pressure and
soil temperature jumps, respectively. In the first example, an unphysical change of 2 hPa
within 1 minute is observed in pressure and in the second example several temperature
spikes are detected (up to 0.6°C change within 1 minute).
The unphysical persistence in time of the measured values are detected by verifying that
the variability within 1 hour is physical, following values in Tab. 3. If the one-minute
values do not vary by more than a specified lower limit (given in Tab. 3) within one
hour, the current value fails the check. Figure 4d shows an example of an unphysical
wind speed measured by a cup anemometer. The value is 0  m/s because of frost
deposition on the sensor and should be compared to the simultaneous unaffected
measurement collected by the sonic anemometer. The persistence test is completed by a
calculation of the standard deviation of temperature, pressure, humidity, and wind
speed for the last one-hour period. In combination with the persistence test, the
evaluation of the standard deviation is a very good tool for the detection of a blocked
sensor as well as a 1-hour sensor drift.

3.3. Specific computation for advanced meteorological variables

The data quality of in-ground temperature and permeability of the soil is checked using
the tests above (Tab. 1 and 3).



The quality of the downwelling shortwave (SW) and longwave (LW) fluxes is tested
following the recommendation of the *Baseline Surface Radiation Network (*BSRN) (Test
version 2.0: Roesch et al, 2011). Additional semi-automatic controls were developed and
applied to SW irradiances in order to reject data collected when the sun-tracker failed
(used for the direct and diffuse SW radiation measurements) and to remove values that
are non-consistent between measured global SW fluxes measured and global SW fluxes
calculated from direct and diffuse measured ones. Individual 1-min native data not
passing the test is automatically removed before performing the 1-hour averages. For
SW fluxes, the global as well as the direct and diffuse irradiance components are
included in the ReOBS dataset. A best estimate of the global SW is calculated as a
combination of the global irradiance measurement and the sum of the diffuse and
horizontal direct irradiance measurements. The sum is taken as default and the blanks
in observations are filled with the global irradiance measurement.
The sensible and latent heat flux data are subjected to spike detection and rejection
algorithms. Sensible and latent heat fluxes are based on sonic measurements and gas
analyzer. The lag between the sonic measurements and the gas analyzer is set to the lag
of maximum correlation over the averaging interval between the sonic anemometer
temperature and the absolute humidity measured by the gas analyzer. At hourly
intervals, sensible and latent heat fluxes are derived from eddy-covariance technics as
well as turbulence statistics. Raw data and calculated statistics are subjected to strict
data limits to reject unphysical values ((13 and (14) in tab. 1). For the latent heat flux,
the open-path InfraRed Gas Analyzer (IRGA) used between 2005 and 2012 could be
damaged by precipitations and was therefore manually switched on and off. The
temporal sampling was thus relatively low and we decided to exchange the IRGA with an
open-path Licor LI-7500 in 2012.






3.4. Retrievals based on remote-sensing measurements developed for ReOBS


3.4.1. Computations for the cloud fraction and cloud base height from lidar

The ReOBS dataset contains Cloud Base Height (CBH) and time series of the Cloud
Fraction (CF), deduced from the SIRTA 355-nm lidar and processed with the STRAT
algorithm (STRucture of the Atmosphere; Morille et al., 2007). The cloud fraction (noted
*cf_nfov,* where "nfov" stands for "narrow field of view") is defined as the number of
profiles containing clouds divided by the total number of profiles collected in one hour.
The cloud base height of the first layer (noted *CBH1*) corresponds to the altitude of the
first cloud layer from the ground as detected by the STRAT algorithm. An hourly cloud
base height is reported in ReOBS only if at least 33% of the profiles collected during this
hour are cloudy and only if less than 40% of the profiles collected during this hour are
noisy (these 33% and 40% thresholds have been chosen based on sensitivity tests in
order to be representative of the selected hour). *CBH2* and *CBH3* respectively are the
altitudes of the base of a second and a third cloud layer (resp.) detected above CBH1 and
separated from the first cloud (the one with CBH1) with clear sky.

3.4.2. The mixing layer depth product

The Mixing Layer Depth (MLD, noted *mld* in SIRTA-ReOBS) is part of the SIRTA-ReOBS
database. It is retrieved from routine lidar measurements (ALS450 from Leopshere
Company) following the method described in Pal et al. (2013) and Haeffelin et al.

(2012).

In this method, the intensity of the lidar-derived aerosol backscatter signal at different
altitudes is used to determine the hourly averaged vertical profiles of variance. Next, the





location of maximum turbulent mixing within the mixing layer is determined and
corresponds to the mean MLD. Micrometeorological measurements of Monin-Obukhov
length scale are used.   (effect of buoyancy on turbulent flow [Monin and Obukhov,
1954])         to         better         determine         the         MLD,         especially
for early morning transition and evening transition periods.  For  these  two  specific
periods,  a first-order approximation on the boundary layer growth rates is obtained and
the variance-based results analysis guides the attribution by searching the altitude of
minimum of the gradient closest to the mean MLD.  Two transition periods of a day are
used  to  distinguish  the  turbulent  regimes  during  the  well-mixed  convective  ABL
(Atmospheric Boundary Layer) and nocturnal/stable MLD.

3.5. Computations for the lidar profiles

The SIRTA ReOBS dataset contains information on the detailed vertical description of
the atmosphere since 2002 from the LNA instrument. A drawback of this instrument is
that it requires human intervention and does not operate when it rains, which
introduces gaps in the data record.
Two different hourly variables are included in the ReOBS dataset:
-   One variable called *STRAThisto*, which contains the number of occurrences of
clear sky, aerosols, clouds, non-valid data, and fully attenuated laser within one
368         hour for each vertical level. The vertical resolution is 15 m up to 15 km and the
layer type classification is based on the STRAT algorithm.

-   One variable called *SRhisto* is a 2D height-intensity number of occurrences
accumulated during one hour, as defined in Chepfer et al. (2010). The lidar signal
intensity is estimated using the scattering ratio SR = ATB/$ATB_{mol}$ where ATB is
the total attenuated backscatter lidar signal and ATBmol is the signal in clear sky



conditions. The vertical resolution is 15 m and the intensity axis contains 18 bins;

375    -999 / -777 / -666 / 0 / 0.01 / 1.2 / 3 / 5 / 7 / 10 / 15 / 20 / 25 / 30 / 40 / 50 /

60 / 80. The value "-999" indicates non-normalized noisy profiles, the value "-
777" is for profiles that cannot be normalized due to the presence of a very low
cloud, and the value "-666" is for non-valid data. ATB profiles are normalized to a
daily molecular profile based on radiosounding measurements launched every
day 10-km away from the SIRTA supersite (at the Météo-France station in
Trappes). The altitude of normalization of ATB (which must be clear sky) is
determined for each profile using the STRAT algorithm.

4. Results

4.1. Description of the ReOBS database content
All data passing the quality control tests are included in the ReOBS final netCDF file. The
variables included in the SIRTA-ReOBS are listed in Tab. 2 together with their
nomenclature (Tab. 2, second column). Figure 5 shows the temporal coverage of each
variable. Some variables such as the classical meteorological variables or the
downwelling radiative fluxes are very well sampled since 2002 when SIRTA activities
started. In contrast, the record for lidar profiles, which started in 2002, contains many
gaps. The sampling of the latent heat flux is much more intermittent than the sampling
of the sensible heat flux due to instrumental issues (see Sect. 3.3).
There are two versions of SIRTA-ReOBS file: a complete file, which includes all
information available (1.2 Gb), and a smaller file, which contains all data except for the
vertical information from the lidar (11.5 Mb). Both data files are available on the



following website: http://sirta.ipsl.fr/reobs.html (tab download, no password required),
which also includes quicklooks and a documentation.
The main added values of ReOBS compared to classical supersite databases are 1) the
vertical profile information coming from lidar measurements, which is user-friendly
thanks to the GOCCP (GCM Oriented CALIPSO Cloud Product) method (Chepfer et al.
201), 2) the possibility to study the troposphere at different time-scales, from daily to
decadal timescales, and 3) the availability of a multi-variable synergetic view of the
atmosphere. And of course a mix of these three aspects. These three main added values
are detailed in the following subsections.

4.2. Vertical profile information

The lidar profiles included in SIRTA-ReOBS provide useful information on the vertical
distribution of clouds and aerosols in the atmosphere. This information together with
many other SIRTA-ReOBS variables have been used recently in various studies (Cheruy
et al. 2012, Chiriaco et al. 2014, Bastin et al. 2016). We first show examples of the two
main ReOBS variables built from lidar measurements (SRhisto and STRAThisto) and we
then describe how these data are used to built cloud fraction profiles. Finally we
describe how to use these data to evaluate clouds simulated by models.

**SRhisto and STRAThisto.** Figure 6 shows *SRhisto* (Fig. 6a) and *STRAThisto* (Fig. 6b) for
every hour containing measurements from 2003 to 2016. Periods without lidar
measurements are not included in this figure and this happens frequently (see Fig. 5)
because measurements are only performed when it is not raining and with human
intervention (i.e. not during night and weekends). Using *SRhisto* the repartition of clouds
can be analyzed as a function of altitude and as a function of the intensity of the lidar



signal (SR), which is a proxy of the cloud optical thickness. Fully attenuated lidar signals
are located in the bin $0 < SR < 1$, clear sky are found in the bin $SR = 1$, uncertain are in
the bin $1 < SR < 5$ (it could be aerosols for instance), and $SR > 5$ is for clouds (white
vertical line in Fig. 6a) (bins defined in section 3.5). The analysis of *SRhisto* shows that
non-precipitating clouds observed at SIRTA (note: the LNA lidar instrument does not
operate when it rains) are mostly thin low clouds (under 4 km with $SR < 15$), or thin
high clouds (above 7 km with $SR < 20$), and there are also thicker clouds with $SR > 80$ or
fully attenuated lidar signal. There are almost no mid-level clouds (between 4 and 7 km)
and only few clouds with $20 < SR < 80$. The analysis of *STRAThisto* indicates that for
these non-precipitating cases, the amount of clouds that fully attenuates the lidar signal
(i.e. "noise") is approximately on the same order of magnitude than the amount of
thinner clouds.

**Cloud fraction profiles.** Cloud fraction profiles are derived from the *SRhisto* or from the
*STRAThisto* variables at a temporal scale ranging from one hour up to several years. At a
given altitude level, the cloud fraction is the ratio between the occurrence of cloudy
cases and the occurrence of all cases excluding the noisy ones. In *STRAThisto* the
occurrence of cloudy layers is given in flag "clouds". In *SRhisto*, a layer is declared cloudy
in a lidar profile when $SR > 5$ and $SR > 1 + \varepsilon / ATB_{mol}$ with $\varepsilon = 1.3 \times 10^{-6}$ SI ($ATB_{mol}$ is
included in SIRTA-reOBS). As expected, the cloud fraction profiles obtained from *SRhisto*
or from *STRAThisto* (Fig. 6d) are different due to the differences in the definition of the
cloud detection in the two algorithms. In particular, *SRhisto* features less low-level
clouds ($z < 4$km) than *STRAThisto*. The magenta curve in Fig. 6c is the SR distribution for
cloudy cases during a given hour for the STRAT algorithm. This distribution shows that
about 28% of these cases correspond to cases where SR cannot be estimated because of



the presence of a very low cloud (-777 in Fig. 6c) preventing the normalization of the
profile (no detection of molecular signal under the cloud). This could explain the
differences of low cloud fractions between *STRAThisto* and *SRhisto* in Fig. 6d. The part of
the magenta curve with values between 0.01 and 5 corresponds to cloudy cases for the
STRAT algorithm but not based on the SR threshold method. This could explain the bias
between CF SR and CF STRAT that occurs at almost all vertical levels. Red and yellow
curves in Fig. 6c also highlight the fact that most of the cases that are defined as PBL or
aerosols for the STRAT algorithm are actually not cloudy when based on the SR
threshold method (the parts of these curves above SR = 5 represent less than 5%). The
differences between STRAT- and SR-based algorithms illustrate the important
sensitivity of the cloud fraction profile to the cloud definition. This sensitivity needs to
be taken into account when comparing the measurements to simulations from GCM or
RCM in order to reproduce the algorithm hypotheses in the simulations: it is usually
done using lidar simulator described below.

**Lidar simulator.** To compare the *SRhisto* variables from SIRTA-ReOBS to GCM or RCM
outputs, we have developed a ground-based lidar simulator, which is similar to the
GOCCP products that have been initially developed for model evaluation, together with
the COSP (CFMIP [Cloud Feedback Model Intercomparison Project] Observation
Simulator Package) lidar simulator (Chepfer et al. 2008). Model outputs are used as
inputs for the lidar simulator to simulate what would be measured from the CALIOP
(Cloud-Aerosol Lidar with Orthogonal Polarization) spatial lidar if the atmosphere were
the simulated one. First the lidar equation that gives the ATB in function of altitude is
used to simulate SR from model outputs. Then the same space and time resolutions as in
observations and the SR thresholds are used for the simulated lidar profiles as in actual



data algorithm (GOCCP for space lidar observations, hist SR for ground-based), making
the lidar profiles directly comparable to the measured ones. In order to use *SRhisto* for
model evaluation in the same way as GOCCP, we have modified the COSP lidar simulator
to make a ground-based lidar version of it. Modifications comparing to the initial version
of the COSP lidar simulator (Chepfer et al. 2008) is the vertical reverse of the lidar
equation following the very first version of lidar simulator described in Chiriaco et al.
(2006). This new version of the ground-based lidar simulator has been used for
comparisons between the SIRTA-ReOBS lidar profiles and the WRF/MED-CORDEX
(Weather Research and Forecast model; Coordinated Regional Climate Downscaling
Experiment for Mediterranean area) simulation in Bastin et al. (2016). This ground-
based version of the lidar simulator is currently implemented to the new COSP2
simulator package (version 2 of COSP, currently developed for CMIP6 simulations),
following these steps: 1) computation of the molecular optical thickness of each layer
(i.e. the atmosphere clear of any particles);, 2) computation of the particles optical
thickness of each layer, 3) computation of the total optical thickness of each layer by
adding the molecular and the particles optical thicknesses, 4) computation of the total
backscatter lidar signal as it would have been measured by a ground-based lidar by
integrating progressively these optical thicknesses from the lowest atmospheric layer to
the top of the atmosphere, and 5) computation of the SR profile by dividing the
attenuated total backscatter lidar profile by the clear sky profile.

4.3. From the daily timescale to the decadal timescale

The temporal variability of the variables included in SIRTA-ReOBS is synthetized in a
single figure, as shown in Fig. 7a for the 2-m temperature.  Each row represents a year
and in each row, the x-axis indicates the day of the year and the y-axis indicates the hour



of the day. This figure allows for the visualization of the presence of gaps in the record
and the different temporal scales of variability: diurnal, seasonal, and interannual. A first
visual inspection leads to the identification of significant anomalies in terms of
amplitude and in terms of persistence. Figure 7b shows the mean temperature diurnal
cycle split by seasons. Solid lines indicate the local SIRTA temperature (-SIR) and dashed
lines indicate the surrounding temperature (-REG, Sect. 3.2.). Since air temperature is at
first order controlled by radiation, the coldest season is winter (mean value 4.1°C)
followed by spring (10.8°C), fall (12°C), and summer (18.5°C), as expected. The
amplitude of the diurnal cycle is greater in summer (standard deviation of 2.7°C), then
spring (STD=2.3°C), fall (STD=1.7°C), and winter (STD=1°C). The specificity of SIRTA
seems to lead to an attenuation of this diurnal cycle, as it is less pronounced than in the
surrounding areas (note that temperatures during daytime are lower at SIRTA than in
the surroundings whereas they are equivalent during the night), likely due to
urban/vegetation/soil moisture effects. Figure 7c shows the mean annual cycle of the 2-
m temperature at 12 UTC (noon; black lines) and at 0 UTC (midnight; grey lines). As for
the diurnal cycle, differences between local SIRTA measurements (solid lines) and
regional 2-m temperature (dashed lines) are more pronounced at noon than at
midnight. Figure 7d shows the interannual variability of the 2-m temperature split into
seasons. There is no significant trend in the four seasons (the linear regression of each of
the four curves multiplied by the number of years is weaker than $1\sigma$(where $\sigma$ is the STD)
of the curve). Nevertheless, significant temperature anomalies are detected such as the
cold winter 2010, the cold spring 2013, the warm fall 2006, the warm winter 2007 or
the hot summer 2003. Summer mean values are split into weather regimes following the
classification of Yiou et al. (2008), which is a deliverable of the A2C2 (Atmospheric flow
Analogues for Climate Change) project. In summer at SIRTA, the daily temperature is



maximal when the weather regime is NAO+ (North Atlantic Oscillation +), it is weaker
when the weather regime is blocking or NAO-, and it is minimal when the weather
regime is "Atlantic Ridge", as expected based on literature (numbers in the box in Fig.
7e). The anomaly (i.e. the mean value of all years is subtracted from each year value) of
$V(y,r_i)$ for June-July-August in a given year $y$ and a given regime $r_i$ (where $r_i$ is one of the
four weather regimes mentioned above) plotted in Fig. 7e is calculated as follows:
$V(y,r_i) = <tas(y,r_i)>/STD(tas(y,r_i))$          (2)
where $<tas(y,r_i)>$ is the mean value of the 2-m air temperature in year $y$ and for days in
regime $r_i$, and STD($tas(y,r_i)$) is its standard deviation. Hence $V(y,r_i)$ is a mean
temperature normalized by its variability and is unitless. Using this estimation, strong
anomalies (i.e. anomalies that have a strong standard deviation) due to only a few
numbers of days are minimized. This representation shows that summers that are not
particularly warm or cold could actually contain significant anomalies. During summer
2013 for instance, NAO- days have been significantly warmer than NAO- days of the
other summers, meaning that during these particular days, temperature anomaly was
due to processes and not only due to the large-scale circulation condition.
Figure 8 is the same as fig. 7 but illustrates the Cloud Radiative Effect (CRE) in the
longwave following the equation:
$CRE_{LW} = rlds - rldscs$          (3)
where $rlds$ and $rldscs$ are the downward all-sky and clear-sky LW flux as defined in Tab.
2. Figure 8a highlights the fact that the database only has few gaps for these variables. It
also shows that the diurnal cycle does not seem to be very intense (about 5 W/m$^2$
amplitude in DJF (December January February) and SON (September October
November), and about 10 W/m$^2$ in JJA (June July August) and MAM (March April May))
whereas the annual cycle is significant (about 25 W/m2 difference between summer and



winter, in particular during the night). Figure 8a-d show that clouds have a stronger
radiative effect in the longwave during winter than during the other seasons regardless
of the hour of the day, for every year. It could simply be due to the amount of clouds that
occur more often during winter, or due to cloud radiative properties that are different
between the seasons. This variable does not have a significant trend from 2003 to 2015
for all season (i.e. the trend is smaller than the standard deviation). Nevertheless, the
mean seasonal values are significantly anti-correlated to the temperature values in
spring (-0.7) and in summer (-0.9). At first order the $CRE_{LW}$ is driven by the amount of
clouds, and the more clouds, the cooler the temperature. This anti-correlation is less
pronounced in winter and fall (-0.5). This is explained by the fact that 1) the
temperature variability must be driven by the air mass circulation more than by clouds,
and that 2) in winter there is less solar radiation even if there are no clouds so the
difference between a clear sky day and a cloudy day is not as pronounced as in summer.
Particular anomalies of $CRE_{LW}$ can be related to the temperature ones: for instance
winter 2007 was particularly mild (fig. 7d) and was associated to weak longwave cloud
radiative effect (fig. 8d) that could be due to a deficit of clouds. On the contrary, winter
2010 was colder than other winters in the period of study and is associated with strong
$CRE_{LW}$. This correlation is also observed in summer (e.g summers 2007 and 2011 are
cold and have strong $CRE_{LW}$). The distinction of $CRE_{LW}$ for each of the four weather
regimes in summer (fig. 8e) shows the part of the $CRE_{LW}$ anomaly that is not due to the
large-scale dynamical conditions, which is the first order driver. The 2013 positive
temperature anomaly for NAO- cases is associated to an important deficit of $CRE_{LW}$ in
this weather regime.

4.4. Multi-variables synergetic view of the atmosphere



One of the main advantages of ReOBS is that all variables are synthetized in a single file
at the same temporal resolution, facilitating studies with multi-variables synergy
particularly useful for the understanding of atmospheric processes. This synergy aspect
has been exploited in previous studies using the SIRTA-ReOBS data, for instance to study
the diurnal cycle, the annual cycle, and the interannual variability but for multiple
variables, (Cheruy et al. (2012) and Bastin et al. (2016)), to study the different
components and scales of the mixing layer depth variability (Pal and Haeffelin, 2015),
and to perform in addition a dynamical analysis (Dione et al. 2016, and Chiriaco et al.

2014).

Figure 9 illustrates a possible synergy of multi-variables. The distribution of three
variables affecting boundary layer processes in summer (JJA) is plotted (colors) as a
function of mixing layer depth (y-axis) and sensible heat flux (x-axis) in the afternoon
(between 2 pm to 6 pm). The occurrence distribution of mixing layer depth versus
sensible heat flux is reported in fig. 9a and then as black contours in each other subplot:
each isoline represents an increment of 0.5%; pixels outside the most external isoline
represent less than 0.5% of the cases (per pixel). The 2-m temperature distribution is
shown on the top figure, the soil moisture at 5-cm depth on the middle one, and the
cloud radiative effect on shortwave fluxes ($CRE_{SW}$) on the bottom one.
Figure 9a shows that shallow boundary layers (altitude of 500-1000 m) in summertime
afternoon are mostly associated with low values of sensible heat flux (0-50 W/m$^2$). They
are associated to strong values of shortwave cloud radiative forcing (<-200 W/m$^2$) due
to the presence of clouds, high soil moisture (> 0.25 g/m$^2$) and low air temperatures (<
17°C). Deeper boundary layers (altitude of 1500-2000 m) are associated with a wide
range of sensible heat fluxes (50-150 W/m$^2$) and generally higher air temperatures (>



22°C). For these deeper boundary layer cases, soil moisture and shortwave cloud
radiative forcing are found to vary significantly.
The role of clouds in the link between *mld* and *hfss* can be easily identified on Fig. 9 In
absence of clouds ($CRE_{SW}$ close to zero), *mld* and *hfss* both have a high amplitude while
they both have a weak amplitude in presence of clouds with strong albedo effect ($CRE_{SW}$
< -200 W/m$^2$). The occurrence of clouds with strong albedo effect correlates well with
low temperatures and high soil moisture values.
However most occurrences (black contours) correspond to low *hfss*, relatively high *mld*,
and intermediate values of $CRE_{SW}$. Temperatures are generally quite high also, and *sm5*
also presents intermediate values. Very clear sky and dry soil conditions ($CRE_{SW}$ > -50
W/m$^2$ and *sm5* < 0.2 g/m$^2$) generally lead to strong sensible heat fluxes and high
temperatures, which do not necessarily translate into higher mixing layer depths than
under cloudier conditions.
In summary, low *mld* are induced by strong cloud albedo effect and thus by low
temperature and weak sensible heat flux due to weak energy reaching the surface. On
the contrary, at hourly time scale, a *mld* higher than 1500 m is associated with a
temperature higher than 20°C and a wide range of $CRE_{SW}$ (although greater than -200
W/m$^2$). But this *mld* can be associated with a weak sensible heat flux. One reason for this
is that the dominant time scale of variability for the boundary layer depth is the daily
timescale, the maximum value being reached generally near 16 UTC in summer above
SIRTA (Pal and Haeffelin, 2015), while the time scale of variability of the boundary layer
forcers is hourly or less (radiative and heat fluxes). The temporal variability around the
*mld* maximal value is often weak during this time lapse because it reacts with a delay.
The energy dissipation rate in the boundary layer is slow and then the boundary layer
stays deep even after the solar energy starts to decrease. So there is a delay between the



decrease of *mld* and the decrease of the sensible heat flux. When considering hourly time
scale, many cases have high *mld* and low *hfss*. Investigating this issue in detail using the
ReOBS database is beyond the scope of this paper.

5. Summary and perspectives
We have presented a set of methods available for the community to robustly process
ground-based data at an hourly time scale over more than a decade. The ReOBS
processing chain has been applied to SIRTA ground-based measurements and leads to
the production of a single netCDF file containing about sixty substantial geophysical
variables hourly averaged over up to a decade. The netCDF file is available at
http://sirta.ipsl.fr/reobs.html    under    http://dx.doi.org/10.14768/4F63BAD4-E6AF-
4101-AD5A-61D4A34620DE .
The main implication of this work is that complex observations are made available for
the scientific community and allow for multiannual and multi-variables studies
combining atmospheric dynamics and thermodynamics, radiation, clouds and aerosols.
For example the variability of 2-m temperature and LW cloud radiative effect can be
jointly studied on the diurnal up to the interannual timescaless. The multi-variables
synergy is also illustrated with a focus on the boundary layer processes. As mentioned
before, SIRTA-ReOBS has been already used in previous published studies: Cheruy et al.
(2012) and Bastin et al. (2016) used SIRTA-ReOBS to evaluate simulations from GCM
and from RCM respectively, and in these studies, using SIRTA-ReOBS has led to identify
the processes responsible of the model biases. Still in term of processes, Pal and
Haeffelin (2015) used SIRTA-ReOBS to study the different components and scales of the
mixing layer depth variability. And Dione et al. (2016) and Chiriaco et al. (2014) have
benefited from SIRTA-ReOBS to study specific season anomalies. Datasets from ReOBS

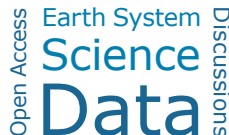

method are also useful tools for teaching and outreach activities such as the European
KIC-Climate summer Journeys of the LABEX L-IPSL (Laboratory of Excellence Institut
Pierre Simon Laplace) CLE-workshop (CLimate and Environment).
The ReOBS processing chain is now complete but the produced files such as SIRTA-
ReOBS are continuously being improved e.g. by adding new periods of data, by treating
new variables, and by improving the quality control. The SIRTA-ReOBS file presented in
this paper is at the time t. Future development for SIRTA-ReOBS include 1) improving
the quality control of classical meteorological variables based on a comparative study of
different methods, 2) adding vertical profiles from radiosounding launched twice a day
10-km away from the SIRTA supersite since the 90's, and 3) adding new variables such
as cloud radar data, gases and wind profiles from radar and lidar.
The ReOBS approach described in this paper will be applied to other supersites.
Applying this approach to data from supersites of the ACTRIS-FR (Aerosol Cloud and
Trace Gases Researche Infrastructure – France) infrastructure, in particular to the P2OA
site located in the South of France is currently being tested. Applying ReOBS to ACTRIS-
EU supersites is also under discussion. Another ongoing project is to integrate the
ReOBS dataset to the OBS4MIP (Observations for Model Intercomparisons Project)
database, which contains the data collected from observations developed specially for
comparisons to CMIP simulations. This requires only few adaptations to fit the OBS4MIP
standards.



**Acknowledgments**
The authors would like to thank the financial support from Ecole Polytechnique, IPSL,
FX-Conseil, and the European project EUCLIPSE (European Union Cloud
Intercomparison, Process Study and Evaluation project) for ReOBS since the beginning
of this project. This study also benefited from the support of the Labex L-IPSL, which is
funded  by the Agence Nationale pour la Recherche (Grant #ANR-10-LABX-0018).
The authors would like to thank IPSL mesocenter and ESPRI teams from IPSL for
providing computing and storage resources, and the SIRTA for providing measurements
and data.
This work is also a contribution to the EECLAT project through LEFE-INSU and CNES
supports, and to ACTRIS-FR which is a national distributed research infrastructure and
identified on the French roadmap for Research Infrastructures, published by the
Ministry of Research. ACTRIS-FR is coordinated by the CONSORTIUM ACTRIS-FR and
comprises a large number of French research organizations and institutions.
Marjolaine Chiriaco was partly supported by Centre National d'Etudes Spatiales (CNES)
until 2016.
The authors would like to thank A2C2 European project for providing the weather
regimes classification (ERC advanced grant No. 338965 - A2C2 ).



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



**List of the tables**

Table 1: List of variables measured at SIRTA and used as inputs for ReOBS.

Table 2: Variables included in SIRTA-ReOBS. First block (family A) is for classical meteorological measurements, second block (family B) is for more advanced measurements, third block (family C) is for parameters retrieved from observations, fourth block (family D) is for vertical lidar measurements.

Table 3: Range of temporal variabilities considered when performing the quality control for the variables listed in the table.



| Measured variable, unity | Instrument | Reference | Physical bounds – *Sensor uncertainty* | Native resolution | Period of obs. |
|---|---|---|---|---|---|
| (1) 2-m air temperature, K | Platinum Resistance Thermometer (PT-100 sensors) | Haeffelin et al., 2005 | -30 /50°C – *0.2°C* | 5sec | 2003-2016 |
| (2) 2-m relative humidity, % | HMP110 hygrometer | | 3/103% - *2%* | 5sec | 2003-2016 |
| (3) Pressure, Pa | PTB110 barometer | | 850/1050hPa – *0.2hPa* | 5sec | 2003-2016 |
| (4) 2-m wind speed, m.s⁻¹ | A100R cup anemometer | | 0/40m.s⁻¹ – *0.2m.s⁻¹* | 5sec | 2003-2016 |
| (5) 2-m wind direction, ° | W200P wind vane | | | | 2003-2016 |
| (6) Precipitation at surface, kg/m²/s | R3070 rain gauge | | 0/50mm.h⁻¹ – *0.1mm* | 5sec | 2003-2016 |
| (7) Surface downwelling LW radiation, W.m⁻² | CG4 or CGR4 pyrgeometers | Ohmura et al., 1998 + BSRN procedures: McArthur, 2004 | 100/500W.m⁻² – *4 W.m⁻²* | 1sec | 2003-2016 |
| (8) Surface downwelling SW radiation, W.m⁻² | Diffuse: Kipp & Zonen CMP22 or CM22 pyranometers Direct: CH1 or CHP1 pyrheliometers | | -5/1200W.m⁻² – *5W.m⁻²* | 1sec | 2003-2016 |
| (9) Surface upwelling LW radiation, W.m⁻² | cg2 30m above ground | | 250/500W/m² – *8 W.m⁻²* | 10sec | 2007-2016 |
| (10) Surface | cm21 | | -5/400W.m⁻² – | 10sec | 2007- |



| upwelling SW radiation, W.m⁻² | 30m above ground | | *10W.m⁻²* | | 2016 |
|---|---|---|---|---|---|
| (11) Soil temperature x[1] cm below ground[1], K | Platinum Resistance Thermometer (PT-100 sensors) | _ | -30/50°C | 5sec | 2007-2016 |
| (12) Soil moisture x[1] cm below ground[1], g/cm³ | Capacitive sensor (ML2x model from Delta-T Devices) | Roth et al., 1992 | 0.05-0.6m3/m3 | 5sec | 2007-2016 |
| (13) 3D wind velocities and virtual air temperature, m/s | METEK (USA-1 standard model) sonic anenometer | Wieser et al., 2001 | 0-30m/s, *0.02m/s* | 10Htz | 2006-2016 |
| (14) water vapor fluctuations, ppt | Open-Path Krypton hygrometer IRGA (Infrared Gas Analyzer) | | 0-60ppt, *2%* | 10Htz | |
| (15) lidar backscattered profiles, - | Leosphere automatic lidar (355 nm) | Haeffelin et al., 2011 | _ | 30sec, 15 m vertical | 2008-2013 |
| (16) lidar backscattered profiles, - | LNA lidar (532 and 1064 nm) | Haeffelin et al. 2005 | _ | 30sec, 15 m vertical | 2003-2016 |
| (17) 360° sky image, - | Yankee Environmental System Total Sky Imager (TSI) | Long et al., 1998 | _ | 1min | 2009-2016 |
| (18) 440 – 870 nm spectral irradiance | Cimel Sunphotometer | Dubovik et al., 2000 | _ | when sun disc is visible | 2008-2016 |
| (19) zenith | GPS | Champolion | _ | 15min | 2008- |





| | | et al. 2004 | | | 2016 |
|---|---|---|---|---|---|
| path delay (ZPD), s | | | | | |
| (20) liquid water path | RPG-HATPRO microwave radiometer | Rose et al., 2005 | _ | 1sec | 2010-2016 |

[1] x is 5 cm, 10 cm, 20 cm, 30 cm, 50 cm

Table 1: List of variables measured at SIRTA and used as inputs for ReOBS.



| | Variable | ReOBS short name | Based on tab. 1 variables | Treatment before ReOBS processing chain |
|---|---|---|---|---|
| A | SIRTA 2-m air temperature, K | tas_SIR | (1) | Direct measurement |
| | SIRTA 2-m relative humidity, % | hurs_SIR | (2) | Direct measurement |
| | SIRTA 2-m specific humidity , kg/kg | huss_SIR | (2) | Simply derived from (2) |
| | SIRTA Sea-level pressure, Pa | psl_SIR | (3) | simply derived from (3) |
| | SIRTA 2-m wind speed, m/s | sfcWind_SIR | (4) | Direct measurement |
| | SIRTA 2-m northward wind, m/s | vas_SIR | (4) (5) | Simply derived from (4) & (5) |
| | SIRTA 2-m eastward wind, m/s | uas_SIR | (4) (5) | Simply derived from (4) & (5) |
| | SIRTA precipitation at surface, $kg/m^2/s$ | pr_SIR | (6) | Direct measurement |
| | Trappes 2-m air temperature, K | tas_TRP | Meteo-FR | Direct measurement |
| | Trappes 2-m northward wind, m/s | vas_TRP | Meteo-FR | Derived from wind speed and direction |
| | Trappes 2-m eastward wind, m/s | uas_TRP | Meteo-FR | Derived from wind speed and direction |
| | Trappes precipitation at surface, $kg/m^2/s$ | pr_TRP | Meteo-FR | Direct measurement |
| B | Surface downwelling LW radiation, $W/m^2$ | rlds | (7) | Direct measurement |
| | Surface downwelling SW radiation, $W/m^2$ | rsds | (8) | Direct measurement |
| | Surface upwelling LW radiation, $W/m^2$ | rlus | (9) | Direct measurement |
| | Surface upwelling SW radiation, $W/m^2$ | rsus | (10) | Direct measurement |
| | Soil temperature x[1] | st$x$[1] | (11) | Direct measurement |



| | | | | |
|---|---|---|---|---|
| | cm bellow ground, K | | | |
| | Soil moisture $x^1$ cm bellow ground, $g/cm^3$ | sm$x^1$ | (12) | Direct measurement |
| C | Lidar cloud fraction | cf_nfov | (15) | Developed for ReOBS based on Morille et al. 2007: Sect. 3.4.1 |
| | Surface downwelling SW radiation for clear sky, $W/m^2$ | rsdscs | (8) | Data parameterization fitting an equation to measured data, accounting zenithal angle, effects of Sun-Earth geometry, mean cloud-free atmospheric components, local surface albedo, subset of measurements error [Dutton et al. 2004] |
| | Surface downwelling LW radiation for clear sky, $W/m^2$ | rldscs | (7) | Analysis of surface irradiance, air temperature, humidity measurements [Long and Turner 2008]; technique with repeatability about 3 W.m$^{-2}$ [Durr and Philipona 2004; Long 2004] |
| | Cloud fraction from sky imager | tot_cld_tsi | | Analysis of color ratio, filtering image into clear or cloudy [Long et al. 1998, 2006] |
| | Cloud fraction from LW radiation | cflw | (7) | APCADA algorithm [Durr and Philipona 2004] |
| | Cloud fraction from SW radiation | cfsw | (8) | Long et al. 2006 |
| | Surface upward sensible, $W/m^2$ | hfss | (13) (14) | Derived from fluctuations of heat and moisture covariances with respect to vertical wind velocity [Brutsaert 1982; Panofsky and Dutton 1984] Variances and covariances rotated to streamwise coordinate for flux computation [Kaimal and Finnigan 1994] |
| | Surface upward latent, $W/m^2$ | hfls | (13) (14) | |



| | | | Corrections for sonic virtual temperature [Schotanus et al. 1983] and density correction for latent heat flux [Webb et al. 1980] |
|---|---|---|---|
| Lidar cloud base height, m | cbh$x^3$ | (17) | Developed for ReOBS based on Morille et al. 2007: Sect. 3.4.1 |
| Aerosol optical thickness at x nm | aot_$x^4$ | (18) | Holben et al. 1998 |
| regional 2-m air temperature, K | tas_REG | Meteo-FR | Developped for ReOBS: Sect. 3.2 |
| regional 2-m northward wind, m/s | vas_REG | Meteo-FR | |
| regional 2-m eastward wind, m/s | uas_REG | Meteo-FR | |
| regional precipitation at surface, kg/m$^2$/s | pr_REG | Meteo-FR | |
| Clear sky integrated water vapor, kg/m$^2$ | water | (18) | Using 675-, 870-, 940-nm channels [Schmid et al. 2001] |
| Aerosol optical thickness at x$^3$ nm | aot_x$^3$ | (18) | Beer-Lambert-Bouguer law [Holben et al. 1998] |
| Angstrom exponent[5] between x$^4$ and y$^4$ nm, nm | x_yangstrom$^4$ | (18) | Eck et al. 1999 |
| Mixing layer depth, m | mld | ?? | Developed in the context of ReOBS: Sect. 3.4.2 [Pal and Haeffelin. 2015] |
| Total GPS water vapor, kg/m$^2$ | iwv | (19) | Businger et al. 1996 |
| Liquid water content, g/m$^2$ | lwp | (20) | Brightness temperature at 23.8 and 31.4 GHz + input from temperature and humidity sensors [Bosisio and Mallet 1998]. Accuracy about 10-20 g.m$^{-2}$ |
| Lidar scattering ratio | SRhisto | (16) | Developed for ReOBS following |



| | | | | |
|---|---|---|---|---|
| | vertical histograms | | | GOCCP method [Chepfer et al. 2010]: Sect. 3.5 |
| D | Lidar STRAT classification vertical histograms | STRAThisto | (16) | Developed for ReOBS applying STRAT algorithm [Morille et al. 2007]: Sect. 3.5 |
| | Lidar molecular profile | Molecular | (16) | Developed for ReOBS applying STRAT algorithm [Morille et al. 2007]: Sect. 3.5 |
| | Altitude of normalization of lidar profiles, m | Alt norm | (16) | Developed for ReOBS applying STRAT algorithm [Morille et al. 2007]: Sect. 3.5 |

[1] x is 5 cm, 10 cm, 20 cm, 30 cm, 50 cm

[2] x is first layer (1), second layer (2), third layer (3)

[3] x is 1020, 870, 675, 500, 440, 380, 340 nm

[4] x and y are the interval between [3] values.

[5] negative slope (or first derivative) of Aerosol Optical Depth (AOD) with wavelength in logarithmic scale is the Angstrom parameter (Eck et al. 1999, see fig. 4r for significance value).

*Table 2: Variables included in SIRTA-ReOBS. First block (category A) is for classical meteorological measurements, second block (category B) is for more advanced measurements, third block (category C) is for parameters retrieved from observations, and fourth block (category D) is for vertical lidar measurements.*



| variable | Temporal variability |
|---|---|
| tas | 5min : ↗ <6°C and ↘ <-9°C<br>60min : ↗ ↘>0.1°C |
| hurs | 5min : ↗ <22% and ↘ <-23%<br>60min : ↗ ↘>0.05% |
| psl | 5min : ↗ <5hPaC and ↘ <-4hPa<br>60min : ↗ ↘>0.1hPa |
| sfcWind | 5min : ↗ <30m/s |
| pr | 5min : ↗ <40mm |
| st$x$[1] | 15min : ↗ <3°C and ↘ <-4°C at -5cm<br>15min : ↗ ↘<3°C at -10cm<br>15min : ↗ ↘<1.5°C at -30cm<br>60min : ↗ ↘>0.05°C |

[1] x is 5 cm, 10 cm, 20 cm, 30 cm, 50 cm

*Table 3: Range of temporal variabilities considered when performing the quality control for the variables listed in the table.*



**Figure captions**

**Figure 1**: Illustration of the routine instruments on the SIRTA supersite.

**Figure 2**: Schematic of the ReOBS general processing chain. Orange, blue, and green boxes and arrows respectively are for steps before, during, and after (resp.) the ReOBS processing chain.

**Figure 3**: (a) Location of the SIRTA supersite and the three neighbouring Météo France stations with their associated weight as defined in the text. Relative occurrence of hourly mean air temperature (b) and wind speed (c) at SIRTA and at the neighbouring Météo France stations between 2005 and 2014, and cumulated precipitation at SIRTA and at the neighbouring Météo France stations in 2012 (d).

**Figure 4**: Example of an unphysical jump in instantaneous values of pressure (a) and temperature in ground at 5 cm (black) and at 10 cm (red) (b). Example of unphysical persistence of high wind speed measurements using with a cup anemometer (d) due to frost (negative temperature in red and high humidity values in blue) (c).

**Figure 5**: Temporal coverage of groups of variables in the SIRTA-ReOBS dataset. In the top panel, blue bars indicate the total numbers of years with data and red bars indicate the mean numbers of days with measurements in a year. In the lower panel, blue bars indicate the numbers of months with data and red bars indicate the mean numbers of hours with measurement in a day. The numbers in brackets are the number of variables in each sub-group. Variables are separated in four categories: classical meteorological measurements (group A, left), more advanced measurements (group B, center-left), variables retrieved from measurements (group C, center-right), and lidar profiles (group D, right). Dn means downward, up means upward.



**Figure 6**: (a) Lidar Scattering Ratio (*SR*) histogram obtained by cumulating all SIRTA observations data from 2003 to 2016. The color bar is the logarithm10 of the percentage of occurrence (the sum of one line is equal to log(100%)), the pink horizontal line corresponds to the altitude of recovery of the lidar (z = 1 km; below this altitude, lidar data is more complicated to use); the white vertical line corresponds to the threshold of cloud detection (*SR* = 5). (b) STRAT histogram obtained by cumulating all data from 2003 to 2016. The color bar is the logarithm10 of the percentage of occurrence. (c) percentage of occurrence of *SR* values for the different STRAT flags (noise in blue – no cases actually -, molecular in green, PBL in red, aerosols in yellow, clouds in magenta, no detection in cyan), cumulating all altitudes above 1 km and only for hours containing a single STRAT flag. (d) Fraction of clouds (in %): *CF SR1* (black solid line) is the occurrence of *SR* > 5 versus the occurrence of SR > 0, *CF SR2* (grey solid line) is the occurrence of *SR* > 5 versus the total occurrence of profiles, *CF STRAT1* (black dashed line) is the occurrence of STRAT cloudy profiles versus the occurrence of STRAT molecular+PBL+aerosols+cloud profiles, *CF STRAT2* (grey dashed line) is the occurrence of STRAT cloudy profiles versus the total occurrence of profiles.

**Figure 7**: Contribution of the multi-temporal scale for the 2-m temperature (in °C). (a) Hourly values, each row corresponds to a year with the day of the year in x-axis, and the hour of the day in y-axis. (b) Mean diurnal cycle averaged from 2003 to 2016 split into seasons (DJF in blue, MAM in green, JJA in red, SON in brown). The mean values and the standard deviation of the 2-m temperature in each season are indicated. (c) Mean annual cycle averaged monthly from 2003 to 2016 at 12 UTC (black line) and at 0 UTC (grey line) with interannual STD in errorbars. (d) Interannual evolution from 2003 to 2016, averaged by season (same colours as (b)). The trends of the curve (i.e. the slope of the curve linear regression multiplied by the number of years – 13) and its standard



deviation are indicated. (e) Same curve as in (d) for JJA only, split per weather regimes (NAO- in cyan, Atlantic ridge in magenta, blocking in green, NAO+ in orange) and plotted in anomaly (i.e. the mean value of all years is subtracted) where "norm. T2" is calculated following the equation (1). In all panels, solid lines are for the SIRTA local 2-m temperature, and dashed lines are for the regional 2-m temperature (around the SIRTA supersite).

**Figure 8**: Same as figure 7 but for the longwave cloud radiative effect (in W/m$^2$). In panel (d), the correlation between values in fig. 7d (tas) and values in fig. 8d ($CRE_{LW}$) are indicated.

**Figure 9**: Occurrence distribution (in %) of the mixing layer depth (y-axis) and the sensible heat flux (x-axis) variables in summer (JJA) for the afternoon (2 pm to 6 pm). Averaged 2-m temperature (b), averaged soil moisture at 5 cm depth (c), and averaged shortwave cloud radiative effect (d). Data are cumulated before the averaging. Black contours in (b), (c) and (d) are isolines of (a): the outermost isoline indicates 0.5% of occurrence and each curve is then incremented of 0.5%, and the internmost curve corresponds to 4%.

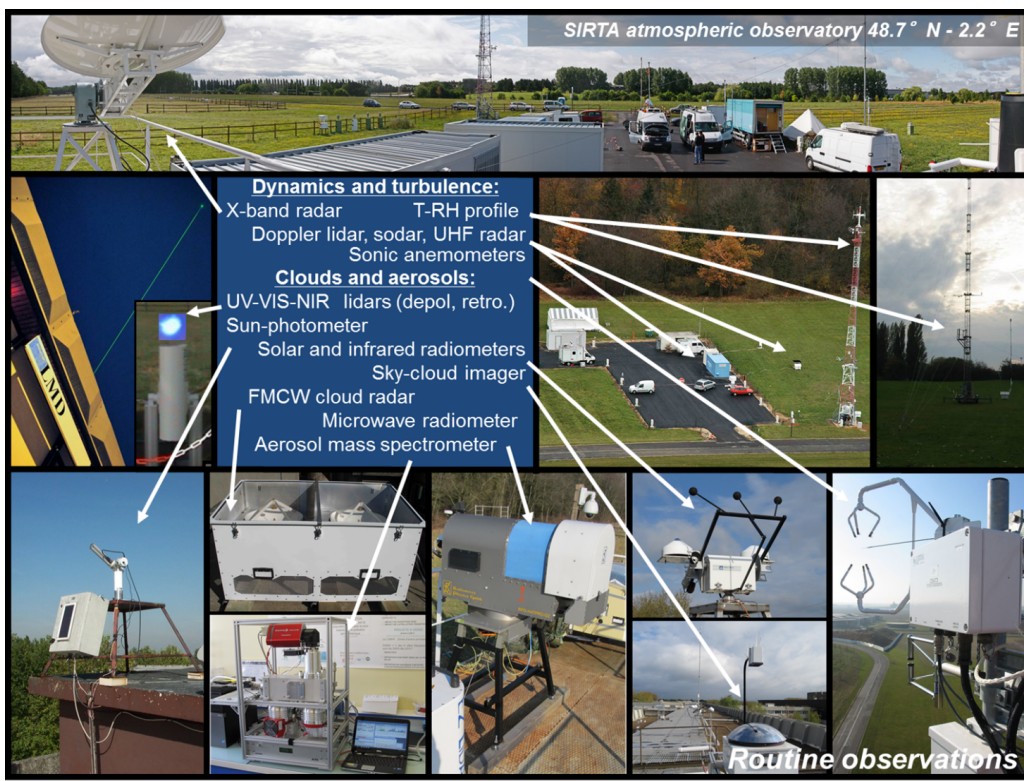

*Figure 1: Illustration of the routine instruments on the SIRTA supersite.*





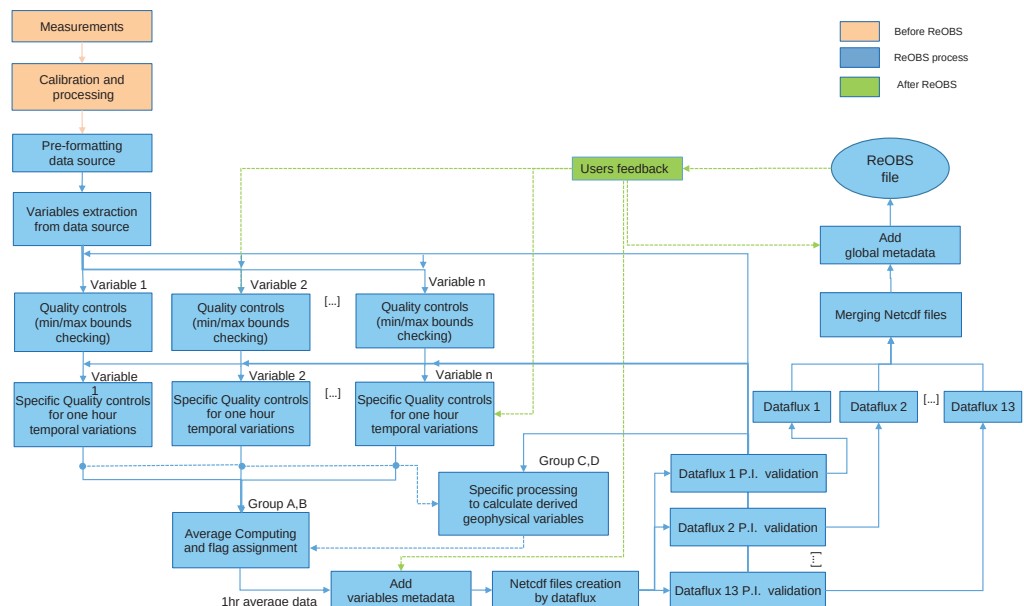

*Figure 2: Schematic of the ReOBS general processing chain. Orange, blue, and green boxes and arrows respectively are for steps before, during, and after (resp.) the ReOBS processing chain.*



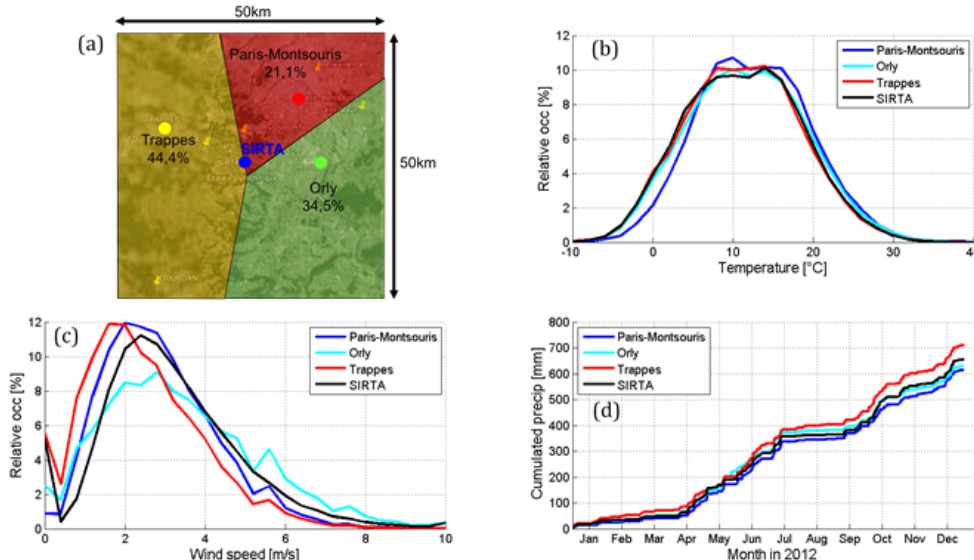

*Figure 3: (a) Location of the SIRTA supersite and the three neighbouring Météo France stations with their associated weight as defined in the text. Relative occurrence of hourly mean air temperature (b) and wind speed (c) at SIRTA and at the neighbouring Météo France stations between 2005 and 2014, and cumulated precipitation at SIRTA and at the neighbouring Météo France stations in 2012 (d).*

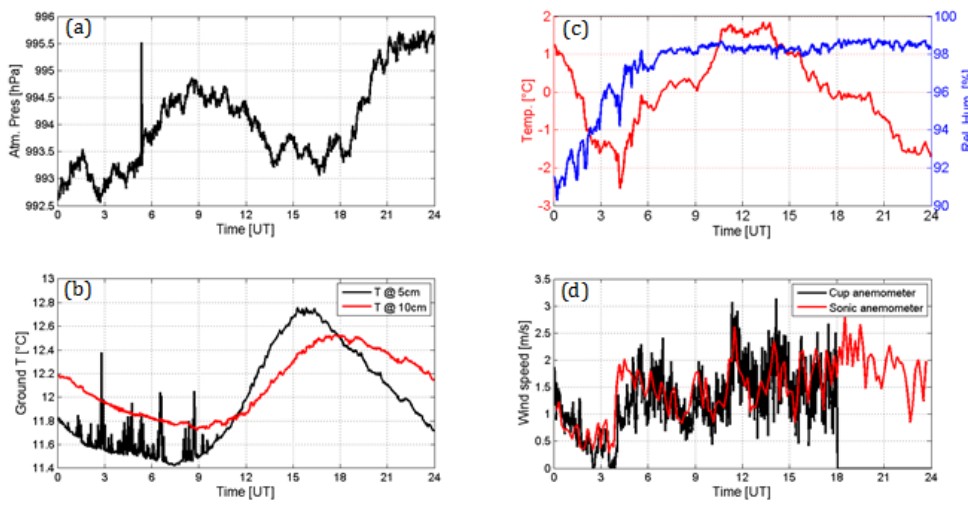

Figure 4: Example of an unphysical jump in instantaneous values of pressure (a) and temperature in ground at 5 cm (black) and at 10 cm (red) (b). Example of unphysical persistence of high wind speed measurements using with a cup anemometer (d) due to frost (negative temperature in red and high humidity values in blue) (c).

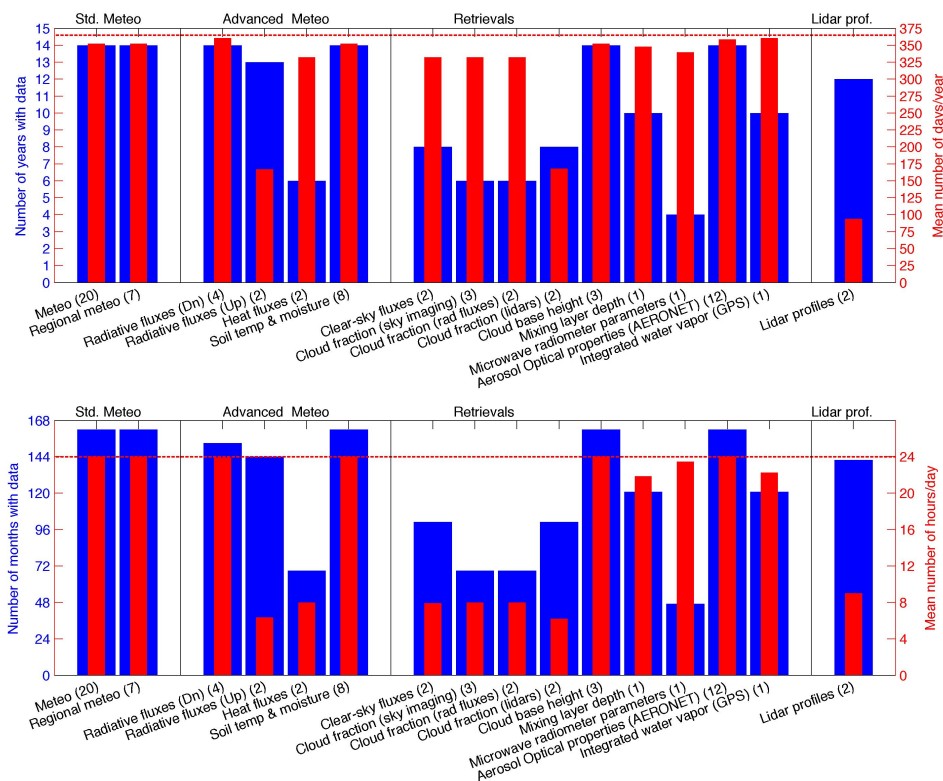

*Figure 5: Temporal coverage of groups of variables in the SIRTA-ReOBS dataset. In the top panel, blue bars indicate the total numbers of years with data and red bars indicate the mean numbers of days with measurements in a year. In the lower panel, blue bars indicate the numbers of months with data and red bars indicate the mean numbers of hours with measurement in a day. The numbers in brackets are the number of variables in each sub-group. Variables are separated in four categories: classical meteorological measurements (group A, left), more advanced measurements (group B, center-left), variables retrieved from measurements (group C, center-right), and lidar profiles (group D, right). Dn means downward, up means upward.*

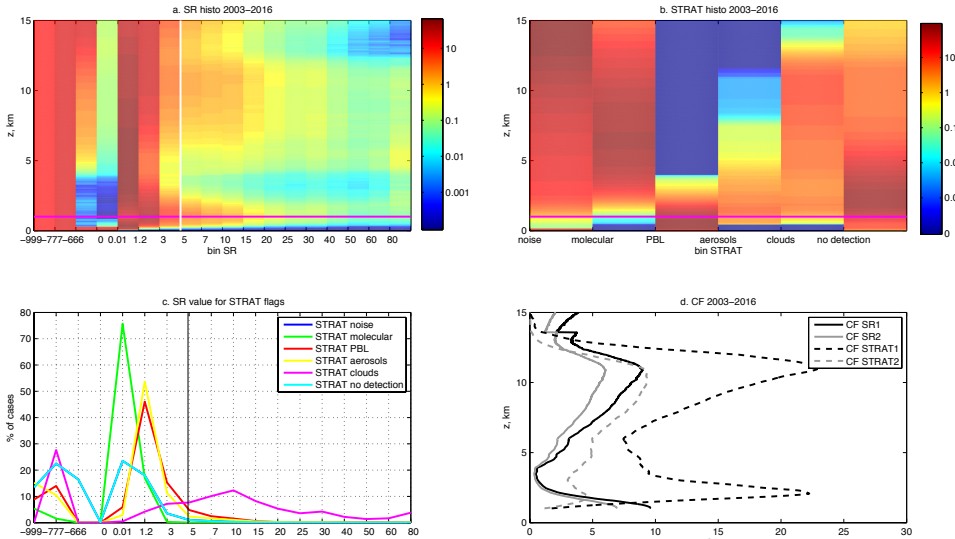

*Figure 6: (a) Lidar Scattering Ratio (SR) histogram obtained by cumulating all SIRTA observations data from 2003 to 2016. The color bar is the logarithm10 of the percentage of occurrence (the sum of one line is equal to log(100%)), the pink horizontal line corresponds to the altitude of recovery of the lidar (z = 1 km; below this altitude, lidar data is more complicated to use); the white vertical line corresponds to the threshold of cloud detection (SR = 5). (b) STRAT histogram obtained by cumulating all data from 2003 to 2016. The color bar is the logarithm10 of the percentage of occurrence. (c) percentage of occurrence of SR values for the different STRAT flags (noise in blue – no cases actually -, molecular in green, PBL in red, aerosols in yellow, clouds in magenta, no detection in cyan), cumulating all altitudes above 1 km and only for hours containing a single STRAT flag. (d) Fraction of clouds (in %): CF SR1 (black solid line) is the occurrence of SR > 5 versus the occurrence of SR > 0, CF SR2 (grey solid line) is the occurrence of SR > 5 versus the total occurrence of profiles, CF STRAT1 (black dashed line) is the occurrence of STRAT cloudy profiles versus the occurrence of STRAT molecular+PBL+aerosols+cloud profiles, CF STRAT1 (grey dashed line) is the occurrence of STRAT cloudy profiles versus the total occurrence of profiles.*



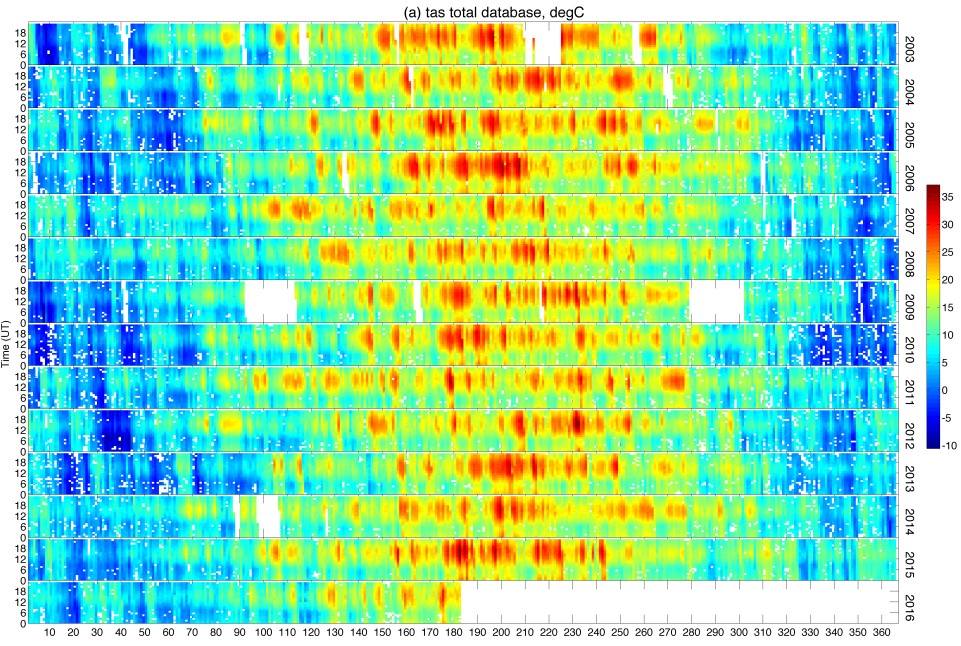

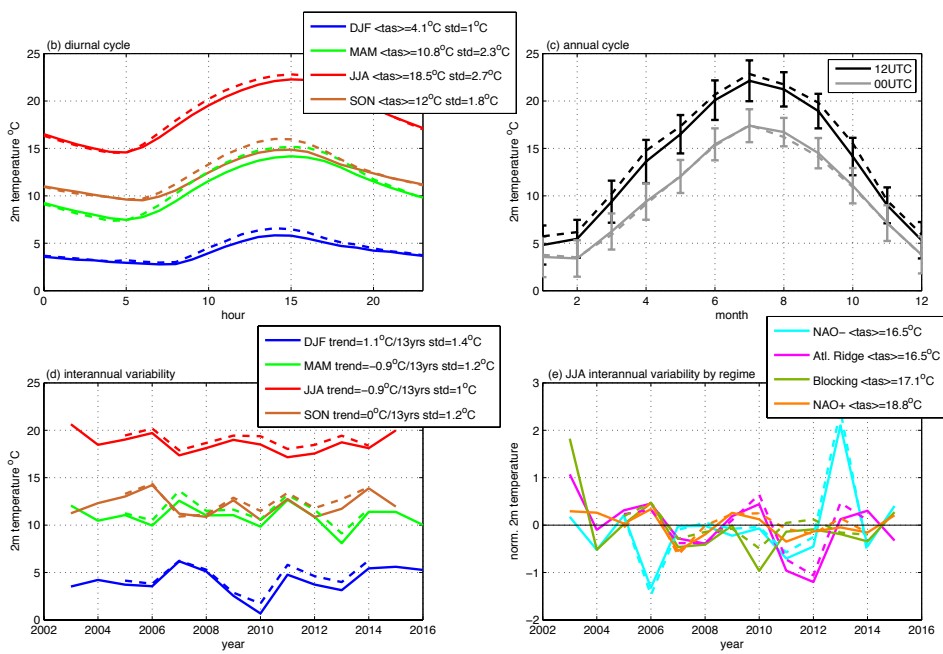

*Figure 7: Contribution of the multi-temporal scale for the 2-m temperature (in °C). (a) Hourly values, each row corresponds to a year with the day of the year in x-axis, and the hour of the day in y-axis. (b) Mean diurnal cycle averaged from 2003 to 2016 split into seasons (DJF in blue, MAM in green, JJA in red, SON in brown). The mean values and the standard deviation of the 2-m temperature in each season are indicated. (c) Mean annual cycle averaged monthly from 2003 to 2016 at 12 UTC (black line) and at 0 UTC (grey line) with interannual STD in errorbars. (d) Interannual evolution from 2003 to 2016, averaged by season (same colours as (b)). The trends of the curve (i.e. the slope of the curve linear regression multiplied by the number of years – 13) and its standard deviation are indicated. (e) Same curve as in (d) for JJA only, split per weather regimes (NAO- in cyan, Atlantic ridge in magenta, blocking in green, NAO+ in orange) and plotted in anomaly (i.e. the mean value of all years is subtracted) where "norm. T2" is calculated following the equation (1). In all panels, solid lines are for the SIRTA local 2-m temperature, and dashed lines are for the regional 2-m temperature (around the SIRTA supersite).*



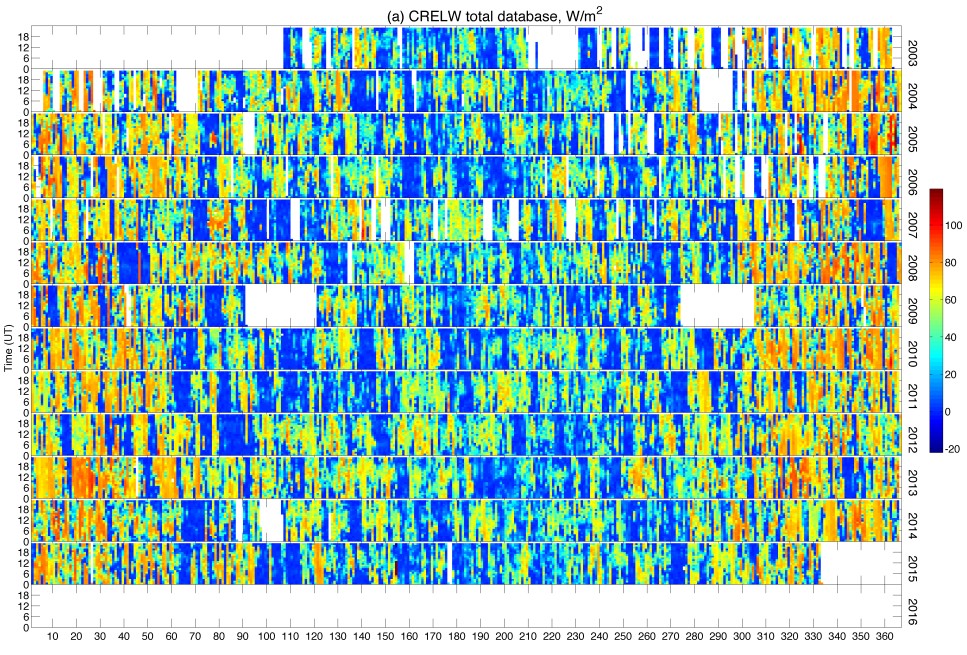



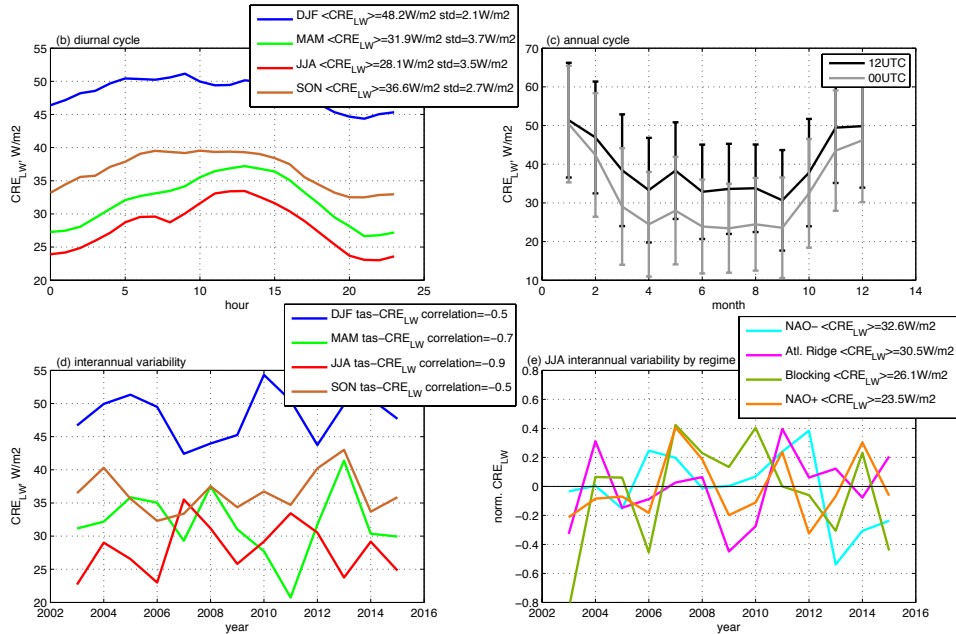

*Figure 8: Same as figure 7 but for the longwave cloud radiative effect (in W/m²). In panel (d), the correlation between values in fig. 7d (tas) and values in fig. 8d (CRE_{LW}) are indicated.*



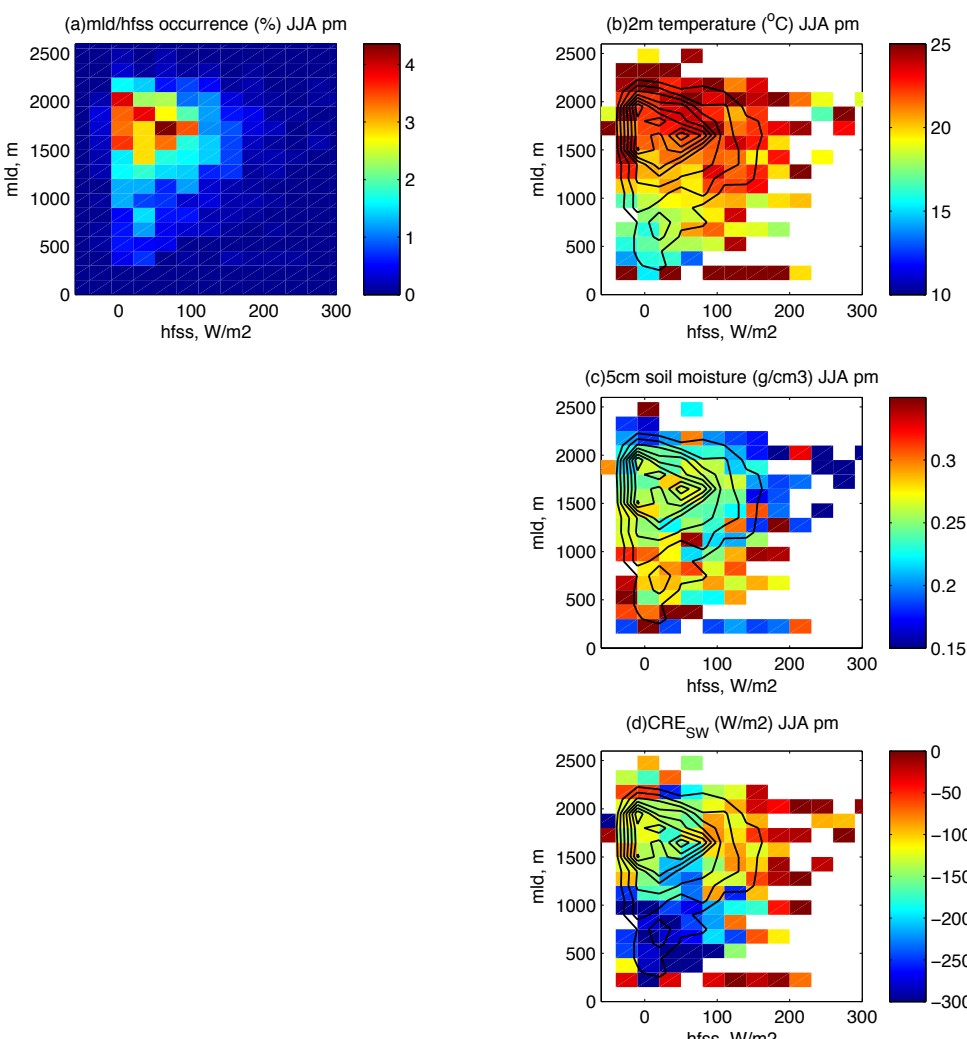

Figure 9: (a) Occurrence distribution (in %) of the mixing layer depth (y-axis) and the sensible heat flux (x-axis) variables in summer (JJA) for the afternoon (2 pm to 6 pm).



*Averaged 2-m temperature (b), averaged soil moisture at 5 cm depth (c), and averaged shortwave cloud radiative effect (d). Data are cumulated before the averaging. Black contours in (b), (c) and (d) are isolines of (a): the outermost isoline indicates 0.5% of occurrence and each curve is then incremented of 0.5%, and the internmost curve corresponds to 4%.*