# Peer review of "ReOBS: a new approach to synthetize long-term multi-variable dataset and"

_Earth System Science Data, 2017_

## Referee Comment (RC1) · Anonymous Referee #1 · 8 Feb 2018

The paper describes data and methods for obtaining a specific and useful dataset from SIRTA supersite observatory. This will foster the use of the data for manifold applications of interest for the whole community. The described methodology could be also of interest for other observational communities for better exploiting and presenting their datasets.

The topic is well suited in the scope of the journal and present high quality data and procedures. There are some small points that could be improved for increasing the impact of the paper itself:

- different time periods are reported in the text, sometimes 15 y, other 14y and often 10

y. Please check this aspect and try to homogenize the wording in this sense

- I counted 40 variables in table 2 and not fifty as stated in the paper. Please check

-page 6, why a decade for the oldest one? you spoke about 15y....confusing for the reader

- page 10: describe the flag 2 or its absence, not clear

- page 11- Figure 3b, 3c and 3d do not illustrate the difference but the pdfs, these plots highlight the eventual differences

- page 11/fig 3c: Orly count seems to be lower in number respect to the others: are the pdf normalized or not? explain this better please.

- page 12: how the weight are assigned? the text here should be clearer

- page 13/table 3 how these numbers are set? Please explain.

- page 15: clearly it is not possible to report here the sensitive tests but please explain in which sense you had this sensitivity test...

- page 16: "the lidar signal intensity is estimated using the scattering ratio" plesae rephrase because literally this is not correct. Additionally, in the Table 1 would be more correct to report Lidar backscattered signals

- -Table 2: you never refer into the text to the Lidar molecular profiles which actually is not clear to me what it is: typically, with lidar measurement molecular profile is not retrieved but assumed from external sources. Please explain better and eventually remove this variable from the dataset.

- page 17: please explain why 2 versions of files are provided by ReOBS

-page 19: line 1: instead of cloud optical thickness I suppose particle optical thickness (i.e. aerosol+cloud) is more suitable

- Fig 6b: the ticks seem to be located not in the middle of corresponding bars but at left

edge, please improve the plots in this sense

- page 20-21 lidar simulator not clear. are these into the reOBS? In any case, would be much better to call it something like space-borne lidar observation simulator, because if I well understood is a tool for simulating the CALIOP signal to be expected if the atmospheric scene observed by SIRTA is captured by the CALIOP.

- Page 20: modify spatial with space-borne

- lines 652-653 probably a cut& paste issue. Please check

- figure 9: report the caption of the color bar close to it and not as title

- table3: please check formatting

-table 1 & 2: the formatting could be improved. I would suggest to have the table horizontally oriented

-line 403: check the reference

---

## Referee Comment (RC2) · Anonymous Referee #2 · 10 Feb 2018

This paper introduces a synthetized long-term multi-variable dataset. It provides a set of methods to process ground-based data at an hourly time scale leading to the ReOBS product for the SIRTA supersite. This process put many variables from various of sources into one single netCDF file and homogenizing them into same format and time resolution, which is very easy for community to use. A webpage link is provided for the data download, quicklooks and documentations. I found the data is well-documented and the paper is well-written. I would also be interested to see the future improvement of including radiosonde profiles. This may be a great value added to the current data. Overall, I think it is a valuable contribution to ESSD with only some minor comments.
Line 31: 'type ob'-> 'types of'

Line 106: move the expansion of ReOBS ("Re" stands for . . .) to line 96.

Line 118 and 123: in other part of the text it is about sixty variables, but here is fifty. Please check the consistency.

Line 260: can you elaborate more on how the given weight corresponds to the geometric representativeness? E.g. a geometry map may be overlapped in Figure 3a.

Line 274-283 and Table 3: this part is kind of confusing. Firstly, I don't understand what the authors exactly mean for the first sentence "as the correlation between the adjacent samples increases with the sampling rate". Are you calculating correlation coefficient for different time windows? According to later part, my understanding is that the non-physical jumps are detected by checking the difference (not the correlation) between measurements in two successive time windows (e.g. 5min), is it right? Secondly, In the limits given in Table 3, what do the upper and lower arrows mean? What do the time windows (5min, 15min, 60min) mean? My guess is that the 60min measurements are used to detect the unphysical persistence by calculating the std. dev. of 1min measurements within this 60min window. Does "5min" mean two successive 5-min measurements are used to test the unphysical jumps? How do the two examples in line 281-283 (2 hPa within 1 minute and 0.6C within 1minute) related with the limits given in Table 3?

Line 288-289: "The value is 0 m/s because of frost deposition on the sensor" -> "The value is 0 m/s after 18 UT because of frost deposition on the sensor (shown by low T and high RH in Figure 4c)"

Line 312-323: is there any corrections imposed on the EC-based fluxes (e.g. density correction (Webb et al. 1980), coordinate rotation (Wilczak et al., 2001), etc)?

References:

Webb EK, GI Pearman, and R Leuning. 1980. "Correction of Flux Measurements for

Density Effects Due to Heat and Water Vapour Transfer." Quarterly Journal of the Royal Meteorology Society 106(44):85-100, doi:10.1002/qj.49710644707.

Wilczak JM, SP Oncley, and SA Stage. 2001. "Sonic Anemometer Tilt Correction Algorithms." Boundary-Layer Meteorology 99(1):127-150, doi:10.1023/A:1018966204465.

Line 337: Can you elaborate what kind of profiles are considered noisy? Does <40% are noisy mean >60% profiles are valid?

Line 351: delete "." After "used"

Line 403: the year might be wrong in "Chepfer et al. 201)"

Line 638: timescaless -> timescales

Line 653: "at the time t" this sentence may not be completed.

Table 1: physical bounds and native resolution are not given for "(5) 2-m wind direction". I think there should be some value, right?

Table 2: some variable short names (e.g. tas_SIR, tas_TRP) are not consistent with the variable names in the downloaded data product.

Figure 4b: there are a lot of spikes and it is not clear which are rejected and which are kept.

Figure 7: "norm. T2" is calculated following the equation (1) -> should be equation (2)
* * *
Comments related to the data (downloaded from the link provided in the manuscript):

1. The current variables are sorted by alphabet order. This is not convenient to find the variables of site information (lat, lon, time, etc). I would suggest moving those variables to the front or to the end, similar to the ARMBE.

2. Since it is a single product with measurements from many instruments, it is important to list the data source (measured from what instrument?) in the attribute of each variable.

3. Some long_names are difficult to understand (e.g., mld, prp, trps, *_l, *_ph). Consider expanding these abbreviations.

4. qc flag: some have flag_meanings in their attributes but some don't

5. std_*: are all these standard deviations within 1-hour time window? Some don't have the "1-hour" in long name.

6. "1-hour std of std_*variable*" should be "1-hour std of *variable*"

7. In some long names of u/v: wing -> wind

8. In global attributes: there are two titles/sources/locations/institutions... and what does the "gps" mean in title/system/source?

---

## Author Comment (AC1) · 4 Apr 2018

**Response to Referee #1**
*Referee comments are in black, responses are in blue.*

- Different time periods are reported in the text, sometimes 15y, other 14y and often 10y. Please check this aspect and try to homogenize the wording in this sense

The ReOBS approach is applied for sets of observations long of at least a decade. SIRTA-ReOBS file cover 15 years. This is now specified in the text.

- I counted 40 variables in table 2 and not fifty as stated in the paper. Please check.

Actually there are 42 lines in Table 2. Nevertheless, some lines refer to several variables: for example, "Soil temperature x[1] cm bellow ground, K" refers to 5 variables (at 5 cm, 10 cm, 20 cm, 30 cm, 50 cm, following the "[1]" note). Finally, there are 64 variables in the file. This is now specified in the text: L119 and L124 "fifty" has been replaced by "sixty". The following sentence has been added L390: "There are 42 lines in Tab. 2, corresponding to 34 variables currently in the file.".

- page 6, why a decade for the oldest one? you spoke about 15y....confusing for the reader

ReOBS is applied to datasets with some variables covering at least a decade, but more if available. For SIRTA-ReOBS, variables with the longest time cover are available over 15 years. This is now specified page 6 (L118 to L125).

- page 10: describe the flag 2 or its absence, not clear

flag 2 is now explained as follow in the text: "- 2: flag 2 is only used for internal control and is never used as an informative output in the ReOBS file", L220.

- page 11- Figure 3b, 3c and 3d do not illustrate the difference but the pdfs, these plots highlight the eventual differences

Following the reviewer comment, the sentence is now: "Figures 3b, 3c, and 3d illustrate the air temperature, wind speed, and cumulated precipitations Probability Density Functions (PDF) at three Météo-France stations within a 50x50km domain around the SIRTA supersite: in Trappes (48.8°N, 2.0°W), in Paris-Montsouris (48.8°N, 2.3°W) and in Orly (48.7°N 2.4°W): these plots highlight the eventual differences from one site to another.".

- page 11/fig 3c: Orly count seems to be lower in number respect to the others: are the pdf normalized or not? explain this better please.

These are relative occurrence, hence the sum of each curve is 100%. The fact that Orly curve seems lower in number must be a visual impression. For wind speed superior to 4.5 m/s, the Orly curve is always above the other ones. It is now specified in L243 that PDF are "in relative occurrence".

- page 12: how the weight are assigned? the text here should be clearer - page 13/table 3 how these numbers are set? Please explain.

The text has been clarified as follow: "A weight is assigned to each of the three stations based on the following method: the 50 x 50 km² domain is divided into $90.10^3$ grid-boxes (300x300), the distance between each box and each site is calculated and then each box is linked to its nearest site. Then then percentage number of boxes linked to each site gives the weight of the site within the domain."

- page 15: clearly it is not possible to report here the sensitive tests but please explain in which sense you had this sensitivity test...

Sensitivity tests based on several case studies have shown that taking less than this 33% or more than this 40% thresholds leads to cloud base height values non-representative of what happens in the current hour. It is now specified in the text L340-342.

- page 16: "the lidar signal intensity is estimated using the scattering ratio" please rephrase because literally this is not correct. Additionally, in the Table 1 would be more correct to report Lidar backscattered signals

The sentence is now "We use the lidar scattering ratio SR...". In table 1, "lidar backscattered profile" has been replaced by "lidar backscattered signal".

- Table 2: you never refer into the text to the Lidar molecular profiles which actually is not clear to me what it is: typically, with lidar measurement molecular profile is not retrieved but assumed from external sources. Please explain better and eventually remove this variable from the dataset.

A paragraph as been added in Sect. 4.2 about that: "Lidar profiles that would be measured in clear sky conditions (so called molecular profiles) is necessary to build SRhisto and STRAThisto as it is used in the SR estimation and in tha STRAT lidar profile normalization. These molecular profiles are estimated based on temperature and pressure profiles measured twice a day by METEO-FRANCE radiosounding at Trappes (10 km from SIRTA). These molecular lidar profiles are included in SIRTA-ReOBS under the *Molecular* variable, as well as the altitude of normalization used for STRAT under the *Alt norm* variable."

- page 17: please explain why 2 versions of files are provided by ReOBS

One version do not contain the lidar vertical profiles so it significantly smaller and then easier to handle. It is now specified L402-403.

- page 19: line 1: instead of cloud optical thickness I suppose particle optical thickness (i.e. aerosol+cloud) is more suitable

cloud optical thickness has been replaced by particle optical thickness.

- Fig 6b: the ticks seem to be located not in the middle of corresponding bars but at left edge, please improve the plots in this sense

The figure has been changed following the reviewer comment.

- page 20-21 lidar simulator not clear. are these into the reOBS? In any case, would be much better to call it something like space-borne lidar observation simulator, because if I well understood is a tool for simulating the CALIOP signal to be expected if the

atmospheric scene observed by SIRTA is captured by the CALIOP.

No it is not into ReOBS, it is into COSP which is a package of different simulators. But, it is done for comparisons with lidar data that are actually into SIRTA-ReOBS (SRhisto). Into COSP, there are two lidar simulators: one is for simulating SR like the CALIOP, one is for simulating SR from a ground-based lidar such as the SIRTA one. This this last simulator which is described in our paper, as it is devoted to comparisons with SRhisto that are in SIRTA-ReOBS. It is now better explained in the "lidar simulator" paragraph page 21:

- Page 20: modify spatial with space-borne

With the reformulation of this paragraph, this expression do not occur anymore.

- lines 652-653 probably a cut& paste issue. Please check

We did not find any mistake on this lines but there was one just above so it is now corrected ("or the" has been added before "CLE-workshop").

- figure 9: report the caption of the color bar close to it and not as title

The figure has been changed following the reviewer comment.

- table3: please check formatting

We had trubles with the marges required by Copernicus, so we will fix that during the proof processes if the paper is accepted.

- table 1 & 2: the formatting could be improved. I would suggest to have the table horizontally oriented

We had trubles with the marges required by Copernicus, so we will fix that during the proff processes if the paper is accepted.

-line 403: check the reference

201 is now replaced by 2010

---

## Author Comment (AC2) · 4 Apr 2018

**Response to Referee #2**
*Referee comments are in black, responses are in blue.*

- Line 31: 'type ob'-> 'types of'

"ob" has been replaced by "of".

- Line 106: move the expansion of ReOBS ("Re" stands for . . .) to line 96.

These precisions L106 have been removed, as it was already explained L97.

- Line 118 and 123: in other part of the text it is about sixty variables, but here is fifty. Please check the consistency.

There are 64 variables in the file. This is now specified in the text: L119 and L124 "fifty" has been replaced by "sixty".

- Line 260: can you elaborate more on how the given weight corresponds to the geometric representativeness? E.g. a geometry map may be overlapped in Figure 3a.

The text has been clarified as follow: "A weight is assigned to each of the three stations based on the following method: the 50 x 50 km$^2$ domain is divided into $90.10^3$ grid-boxes (300x300), the distance between each box and each site is calculated and then each box is linked to its nearest site. Then then percentage number of boxes linked to each site gives the weight of the site within the domain."

- Line 274-283 and Table 3: this part is kind of confusing. Firstly, I don't understand what the authors exactly mean for the first sentence "as the correlation between the adjacent samples increases with the sampling rate". Are you calculating correlation coefficient for different time windows? According to later part, my understanding is that the non-physical jumps are detected by checking the difference (not the correlation) between measurements in two successive time windows (e.g. 5min), is it right? Secondly, In the limits given in Table 3, what do the upper and lower arrows mean? What do the time windows (5min, 15min, 60min) mean? My guess is that the 60min measurements are used to detect the unphysical persistence by calculating the std. dev. of 1min measurements within this 60min window. Does "5min" mean two successive 5-min measurements are used to test the unphysical jumps? How do the two examples in line 281-283 (2 hPa within 1 minute and 0.6C within 1minute) related with the limits given in Table 3?

"as the correlation... rate": this sentence has been removed, as actually, jump detections are not based on correlations. In Table 3, upper arrow means an increase during the time window indicated, and lower arrow means a decrease during the time window indicated. It is now specified in the table caption. All these tests are done with the variables native resolution which is 5 seconds for measurements (1) to (6) and (11) and (12) (Table 1). It is now specified L276.

For the two examples concerning pressure and soil temperature, the measured temporal variability for 1min time resolution is larger than the possible range of variability shown in table 3, i.e. 1hPa and 0.2°C, respectively. In this case the corresponding variables are

not accounted for. The text has been reworked to precise as follow: "In the first example, an unphysical change of 2 hPa within 1 minute is observed in pressure (larger than 5hPa during 5min, see table 3). In the second example several temperature spikes (0.6°C within 1min for ground at -5cm) are detected and we reject the data when the increase reaches +3°C and the decrease -4°C within 15min (i.e +0.2°C and -0.27°C for 1min resolution)."

- Line 288-289: "The value is 0 m/s because of frost deposition on the sensor" -> "The value is 0 m/s after 18 UT because of frost deposition on the sensor (shown by low T and high RH in Figure 4c)"

The sentence has been modified following the reviewer comment.

- Line 312-323: is there any corrections imposed on the EC-based fluxes (e.g. density correction (Webb et al. 1980), coordinate rotation (Wilczak et al., 2001), etc)?

Webb EK, GI Pearman, and R Leuning. 1980. "Correction of Flux Measurements for Density Effects Due to Heat and Water Vapour Transfer." Quarterly Journal of the Royal Meteorology Society 106(44):85-100, doi:10.1002/qj.49710644707.

Wilczak JM, SP Oncley, and SA Stage. 2001. "Sonic Anemometer Tilt Correction Algorithms." Boundary-Layer Meteorology 99(1):127-150, doi:10.1023/A:1018966204465.

The reviewer is right, we had some details as follow : "with the open-path InfraRed Gaz Analyser the molar density fluctuations are accounted for in the processing by following the classic formulation of Webb et al. (1980). Moreover, automatic method has been applied to correct wind statistics for any misalignment of the sonic anemometer with respect to the local wind streamlines of the sonic anemometer with respect to the local wind streamlines according to Wilczak et al. (2001).".

- Line 337: Can you elaborate what kind of profiles are considered noisy? Does <40% are noisy mean >60% profiles are valid?

A bracket has been added at the end of the sentence to precise that actually it means that "i.e. at least 60% of profiles are valid".

- Line 351: delete "." After "used"

The dot has been deleted following reviewer comment.

- Line 403: the year might be wrong in "Chepfer et al. 201)"

201 is now replaced by 2010.

- Line 638: timescaless -> timescales

timescales has been replaced by timescales.

- Line 653: "at the time t" this sentence may not be completed.

"at the time t" has been replaced by "at a precise time".

- Table 1: physical bounds and native resolution are not given for "(5) 2-m wind

direction". I think there should be some value, right?

The native resolution is 5 sec, and the physical bounds are 0 and 360 degrees. It is now specified in the table.

- Table 2: some variable short names (e.g. tas_SIR, tas_TRP) are not consistent with the variable names in the downloaded data product.

This is right. It has been corrected in the table and in the text: suffix are actually -sirta, -regional, and -trps.

- Figure 4b: there are a lot of spikes and it is not clear which are rejected and which are kept.

As suggested we had some details as follow : "In the second example several temperature spikes  (ground at -5cm) are detected and we reject the data when the increase reach +3°C and the decrease -4°C within 15min."

- Figure 7: "norm. T2" is calculated following the equation (1) -> should be equation (2)

equation (2) is now indicated instead of equation (1).

- Comments related to the data (downloaded from the link provided in the manuscript):

1. The current variables are sorted by alphabet order. This is not convenient to find the variables of site information (lat, lon, time, etc). I would suggest moving those variables to the front or to the end, similar to the ARMBE.

It is true. We make the commitment that it will be done in the next version of the production file, as it is done every 6 months.

2. Since it is a single product with measurements from many instruments, it is important to list the data source (measured from what instrument?) in the attribute of each variable.

As it is explained in the paper, since the sources could be multiple for one single variable, we made the choice to not explain the instrument in the attribute of each variable. To get this information, it is then necessary to refer to the documentation, and to the present paper (Table 1).

3. Some long_names are difficult to understand (e.g., mld, prp, trps, *_l, *_ph). Consider expanding these abbreviations.

It is a good idea: we make the commitment that it will be done in the next version of the production file, as it is done every 6 months.

4. qc flag: some have flag_meanings in their attributes but some don't

The attribute "flag_meaning" has been added to all qc variables.

5. std_*: are all these standard deviations within 1-hour time window? Some don't have the "1-hour" in long name.

All std variables are actually the standard deviation estimated over one hour. It is now precised in all std variables attributes.

6. "1-hour std of std_*variable*" should be "1-hour std of *variable*"

This has been corrected following the reviewer comment.

7. In some long names of u/v: wing -> wind

This has been corrected.

8. In global attributes: there are two titles/sources/locations/institutions... and what does the "gps" mean in title/system/source?

It was a bug, thank you. It has been corrected.